# LAYER-WISE UNIVERSAL APPROXIMATION AND PROGRESSIVE OPTIMIZATION FOR RESIDUAL NETWORKS

## ABSTRACT

This paper provides a theoretical analysis to bridge the gap wherein the classical UAT, originally developed for feedforward networks (FNNs), fails to directly apply to modern residual networks (RNs) like ResNets and Transformers. Our key contributions are: First, we prove a layer-wise UAT for residual networks by formulating ResNet and Transformer blocks in a unified form compatible with FNNs, ensuring that each layer satisfies the UAT conditions (compact inputs and continuity). Second, using this layer-wise formulation, we demonstrate that RN training can be effectively modeled as a compensatory additive model, enabling sequential optimization where layers collaboratively reduce input-output divergence. Unlike conventional end-to-end training (which suffers from instability risks), our layer-wise approach ensures freedom constrained learning and superior convergence. Third, we propose Layer-wise Progressive Approximation (LPA), a training paradigm that enforces convergence in early layers to faciliate convergence in later ones. Experimental results across both synthetic datasets and standard benchmarks (CIFAR-10/100, Fashion-MNIST) demonstrate LPA's significant advantages: up to 8.31% higher accuracy alongside early-layer convergence and improved training stability compared to conventional end-to-end approaches. We further show that adding an adaptive criterion to LPA automatically discovers the effective layers and prunes the remainder, enabling aggressive compression, reducing model size by up to 79.17%. This suggests that simply scaling up the model is often unnecessary. The source code will be released unpon acceptance at https://(open_upon_acceptance).

## 1 INTRODUCTION

Research on the function approximation abilities of deep neural networks (DNNs) has largely centered on the Universal Approximation Theorem (UAT) Cybenko (1989); Hornik et al. (1989); Hornik (1991). A key limitation remains: classical UAT results were derived for feedforward neural networks (FNNs) and do not directly extend to modern architectures with varied operations and residual connections, such as ResNets (He et al., 2015) and Transformers (Vaswani et al., 2017). This theoretical gap is increasingly concerning, given that most contemporary DNNs are residual networks.

Recent foundational work by Lin & Jegelka (2018) demonstrated the universal approximation properties of deep ResNets, with Yun et al. (2019a); Kratsios et al. (2021) establishing similar results for Transformers. However, these results are architecture-specific, making the conclusions hard to generalize across different residual network families. This paper closes that gap by formulating a unified approximation framework with three key findings/contributions:

**UAT is feasible to RNs at layer-level**: We prove that both ResNet and Transformer blocks can be unified into a layer-level representation compatible with FNNs. We demonstrate that standard UAT requirements, such as input compactness and target function continuity, can be satisfied at each layer, establishing the mathematical foundation for layer-wise UAT.

**RNs can be learned with progressive approximations**: Through inductive analysis of inter-layer dynamics, we model RN learning as a compensatory additive problem where layers collaboratively reduce input-output divergence. We reveal that conventional end-to-end training resembles simultaneous optimization (Betts, 2009; Saad, 2003; Nocedal & Wright, 2018), where excessive degrees of

freedom in early layers create oscillation risks. We demonstrate that if layer learning begins only after its predecessor layers have been properly trained, the layers operate in a progressive optimization manner that can reduce input-output divergence more effectively and reduce oscillation risk.

**Practical implementation via LPA**: Building the theoretical formulation, we propose LPA, which trains RNs progressively so that enforcing convergence in early layers facilitates convergence in later ones. This stabilizes training and helps avoid local minima. Our experimental results on both synthetic and real datasets validate LPA's advantages with stable convergence and improved accuracy (up to 8.31% performance gain on CIFAR-10, CIFAR-100 (Krizhevsky, 2009), and Fashion-MNIST (Xiao et al., 2017) over end-to-end learning). In addition, our adaptive LPA training compresses model size by up to 79.17%, suggesting that simply scaling up is often unnecessary.

## 2 RELATED WROK

**Universal Approximation Theorem:** The study of neural networks' approximation power began with Cybenko (1989), who formulated UAT and proved the universal approximation property for single-hidden-layer fully connected networks. (In the following text, when we refer to UAT, we specifically mean the mathematical formulation provided by Cybenko (1989).) Hornik et al. (1989); Hornik (1991) extended it to a multilayer network and proved a multilayer network can approximate any Borel measureable function, showing that any measurable function between finite-dimensional spaces can be approximated. Since then, most theoretical work has been built around the UAT.

Early neural architectures were mainly FNNs (Rumelhart et al., 1986) and plain convolutional neural networks (CNNs) (LeCun et al., 1998; Krizhevsky et al., 2012), both of which can be recast as multi-layer fully connected networks and thus satisfy UAT conditions. Later, ResNets were introduced by He et al. (2015) and gained popularity for their strong performance and ability to mitigate gradient vanishing (Philipp et al., 2017). Some works (Yun et al., 2019b; Chen et al., 2021) specifically study the representation capabilities of ResNets. This evolution paved the way for more complex architectures, including Inception-v4 Szegedy et al. (2017), the Transformer Vaswani et al. (2017), BERT Devlin et al. (2019), and ViT Dosovitskiy et al. (2020). Meanwhile, analyzing their approximation capacity is more challenging, since residual architectures do not fit the standard FNN framework.

**Universal Approximation Ability of Residual Architectures :** Many efforts have been made to tackle this challenge. For example, Hardt & Ma (2016); Lin & Jegelka (2018) show that architectures with an FCN core retain universal approximation capability when residual links are added; likewise, Oono & Suzuki (2019); He et al. (2021) have established that CNNs augmented with residual links also satisfy universal approximation properties. Moreover, Transformers have been shown to possess universal approximation capabilities as well Yun et al. (2019a); Kratsios et al. (2021).

Despite encouraging progress, most prior results are architecture-specific: they turn different architecture into corresponding specific formula and adapt various theorems or techniques tailored to particular models (e.g., Kolmogorov's theorem (Jiang & Li, 2024; Jiao et al., 2025) and the function space density (Lin & Jegelka, 2018)). These ad-hoc approaches are difficult to generalize across different residual architectures. This work addresses that gap by providing a unified theoretical framework, supported by experiments on both ResNets and Transformers.

To this end, we argue that previous works have primarily focused on architecture-specific proofs, and there have not yet been an attempt to draw conclusions that are generalizable to the broader family of residual networks. This approach increases analytical complexity and typically necessitates additional assumptions or constraints (e.g., Kajitsuka & Sato (2023); Lin & Jegelka (2018)). In contrast, we take a layer-wise perspective. We unify representations of architectures with different cores (e.g., ResNets and Transformers), establish UAT properties at the layer level, and then analyze the full architecture via inter-layer interactions.

# 3 LAYER-WISE UAT FOR RESIDUDAL NETWORKS

## 3.1 UNIVERSAL APPROXIMATION THEOREM

The UAT (Cybenko, 1989) is one of the fundamental and widely acknowledged theoretical corner-stones in the field of deep learning. Although its understanding has evolved over time, the central mathematical formulation has remained unchanged. Let $\sigma$ denote any continuous sigmoid activation function. A feedforward neural network (FNN) can then be expressed as a finite sum in the form:

$$G(\mathbf{x}) = \mathbf{W}'\sigma\left(\mathbf{W}\mathbf{x} + \mathbf{b}\right) + \mathbf{b}', \tag{1}$$

where the weight matrices $\mathbf{W}'$ and $\mathbf{W}$, and bias vectors $\mathbf{b}$ and $\mathbf{b}'$ are learnable.

**Universal Approximation Theorem** – *For any continuous function ($f \in C(\mathbf{I}_n)$) which is defined in a compact domain $\mathbf{I}_n$) and any $\varepsilon > 0$, there exists a function $G(\mathbf{x})$ such that:*

$$\sup_{\mathbf{x} \in \mathbf{I}_n} \left\| G(\mathbf{x}) - f(\mathbf{x}) \right\| < \varepsilon, \tag{2}$$

*indicating that, when the number of neurons in the hidden layer is sufficiently large, FNNs can approximate any continuous function on a closed interval to an arbitrary degree of precision.*

## 3.2 CONNECTING RESIDUAL NETWORKS TO UAT

We adopt a layer-wise analytical approach to establish a connection between their structures and thus UAT can be applied. A layer $i$ of an RN can be expressed as:

$$\mathbf{x}_i = \mathbf{x}_{i-1} + \mathbf{W}'_i\sigma(\mathbf{W}_i\mathbf{x}_{i-1} + \mathbf{b}_i) + \mathbf{b}'_i, \tag{3}$$

where $\mathbf{x}_{i-1}$ and $\mathbf{x}_i$ represent the input and output of the $i$-th layer, respectively. This recursive formulation is specific to residual networks and does not align with the conventional form of FNNs. To address this, let us redefine the residual mapping as:

$$G_k(\mathbf{x}_{k-1}) = \mathbf{W}'_k\sigma(\mathbf{W}_k\mathbf{x}_{k-1} + \mathbf{b}_k) + \mathbf{b}'_k, \tag{4}$$

which is itself a FNN. Substituting this into Eq. (3), we can rewrite the layer formulation as:

$$\mathbf{x}_i = \mathbf{x}_{i-1} + G_k(\mathbf{x}_{k-1}). \tag{5}$$

**Single Layer RN**: Let $f : K_0 \to \mathbb{R}^d$ represent a continuous target function, and the input space $K_0 \subset \mathbb{R}^d$ be a compact set. Let us examine the simplest case of a single-layer RN, expressed as:

$$\mathbf{x}_1 = \mathbf{x}_0 + G_1(\mathbf{x}_0), \quad \mathbf{x}_0 \in K_0, \tag{6}$$

where the learning goal is to minimize the approximation error:

$$G_1^*(\mathbf{x}_0) = \arg\min_{G_1} \left\| f(\mathbf{x}_0) - \mathbf{x}_1 \right\| = \arg\min_{G_1} \left\| f(\mathbf{x}_0) - \mathbf{x}_0 - G_1(\mathbf{x}_0) \right\| = \arg\min_{G_1} \left\| f_1(\mathbf{x}_0) - G_1(\mathbf{x}_0) \right\|,$$

with $f_1(\mathbf{x}_0) = f(\mathbf{x}_0) - \mathbf{x}_0$ being the transformed target function. It reduces the problem to learn a FNN $G_1^*(\mathbf{x}_0)$ that approximates the continuous function $f_1(\mathbf{x}_0)$. By the UAT, there exists a $G_1^*(\mathbf{x}_0)$ satisfying:

$$\sup_{\mathbf{x}_0 \in K_0} \left| G_1^*(\mathbf{x}_0) - f_1(\mathbf{x}_0) \right| < \varepsilon. \tag{7}$$

**Two-Layer RN**: Now, consider adding a second layer to the RN. The output is expressed as:

$$\mathbf{x}_2 = \mathbf{x}_1 + G_2(\mathbf{x}_1), \quad \mathbf{x}_1 \in K_1. \tag{8}$$

Here, the space of $\mathbf{x}_1$ is given by the Minkowski sum:

$$K_1 := K_0 + G_1(K_0) = \{\mathbf{x}_0 + G_1(\mathbf{x}_0) \mid \mathbf{x}_0 \in K_0\}. \tag{9}$$

Since $K_0$ is compact and $G_1$ is continuous, $G_1(K_0)$ is compact (as continuous mappings preserve compactness). Thus, the Minkowski sum $K_1$ remains compact. The learning goal becomes:

$$G_2^*(\mathbf{x}_1) = \arg\min_{G_2} \left\| f(\mathbf{x}_0) - \mathbf{x}_2 \right\| = \arg\min_{G_2} \left\| f(\mathbf{x}_0) - \mathbf{x}_1 - G_2(\mathbf{x}_1) \right\| = \arg\min_{G_2} \left\| f_2(\mathbf{x}_0) - G_2(\mathbf{x}_1) \right\|,$$

with $f_2(\mathbf{x}_0) = f(\mathbf{x}_0) - \mathbf{x}_0 - G_1(\mathbf{x}_0)$. Assuming $G_1^*(\mathbf{x}_0)$ is already learned for $G_1(\mathbf{x}_0)$, $\mathbf{x}_1$ becomes fixed, reducing the problem to learning $G_2^*(\mathbf{x}_1)$, a FNN approximating $f_2(\mathbf{x}_0)$. By UAT, the existence of $G_2^*(\mathbf{x}_1)$ is guaranteed and $G_2^*(\mathbf{x}_1)$ shound satify Factorization Continuity Theorem (see Appendix G).

**Generalizing to N Layers**: For an $N$-layer RN, the output is given by:

$$\mathbf{x}_N = \mathbf{x}_{N-1} + G_N(\mathbf{x}_{N-1}), \tag{10}$$

and the learning goal can be generalized as:

$$G_N^*(\mathbf{x}_{N-1}) = \arg\min_{G_N} \big\| f_N(\mathbf{x}_0) - G_N(\mathbf{x}_{N-1}) \big\|, \tag{11}$$

$$f_N(\mathbf{x}_0) = f(\mathbf{x}_0) - \mathbf{x}_0 - \sum_{i=1}^{N-1} G_i(\mathbf{x}_{i-1}), \tag{12}$$

where $f_N(\mathbf{x}_0)$ is the transformed target function for the $N$-th layer. By applying UAT to each layer, the existence of $G_N^*(\mathbf{x}_{N-1})$ is ensured. This bottom-up analysis highlights that residual networks can be understood as iterative FNNs where each layer incrementally approximates a transformed target function, with UAT guaranteeing the feasibility of this approach.

## 3.3 SIMULTANEOUS VS. SEQUENTIAL LEARNING PARADIGMS

From Eq. (12), the loss function of an N-layer RN can be written as:

$$\mathcal{L}(\theta_1, \theta_2, \cdots, \theta_N) = f(\mathbf{x}_0) - \mathbf{x}_0 - G_1(\mathbf{x}_0) - G_2(\mathbf{x}_1) - G_3(\mathbf{x}_2) - \cdots - G_N(\mathbf{x}_{N-1}), \tag{13}$$

where $\theta_i$ is the set of parameters that defines $G_i()$ (i.e., the FNN at the $i^{th}$ layer). Eq. (13) is a compensatory additive system, with $f(\mathbf{x}_0) - \mathbf{x}_0$ fixed as the input-output divergence. The learning objective of the N-layer RN is to optimize the N FNNs (i.e., $\{\theta_i\}$) and minimize this divergence.

In numerical linear algebra and control/optimization, two prevalent paradigms are Simultaneous (all-at-once) and Sequential optimization (Betts, 2009; Saad, 2003; Nocedal & Wright, 2018). In Simultaneous optimization, N layers work collaboratively with their parameters updated jointly as

$$(\theta_1, \theta_2, \cdots, \theta_N)^* = \arg\min_{\theta_1, \theta_2, \cdots, \theta_N} \mathcal{L}(\theta_1, \theta_2, \cdots, \theta_N), \tag{14}$$

which is exactly the end-to-end learning paradigm commonly adopted for RNs.

Although sequential optimization is rarely employed in modern deep learning systems, it remains theoretically appealing due to its conceptual alignment with the layer-wise UAT conditions. For analytical purposes, we assume such an optimization procedure could be implemented (i.e., training the $N$ layers of a FNN sequentially where each layer optimizes its parameters based on the output of the preceding layer before passing the residual to the next.) In other words, layer $G_i$ minimizes the input-output discrepancy to the best of its ability, after which $G_{i+1}$ further reduces the remaining discrepancy. This learning process is formally expressed as:

$$(\theta_1, \theta_2, \cdots, \theta_N)^* = \arg\min_{\theta_N} \cdots \arg\min_{\theta_2} \arg\min_{\theta_1} \mathcal{L}(\theta_1, \theta_2, \cdots, \theta_N). \tag{15}$$

Sequential seems to have two advantage in form. First, it closely parallels our layer-wise UAT analysis: once the $i^{th}$ layer is optimized, it effectively clarifies the learning objective for the $(i+1)^{th}$ layer. This is harder to satisfy under simultaneous training. Second, Sequential optimization may achieve faster convergence, as the optimization of the $i^{th}$ layer constrains the degrees of freedom of the parameter space for the $(i+1)^{th}$ layer, thereby reducing the oscillation risk of the learning process for subsequent layers. However, the strong inter-layer dependencies in Sequential optimization can cause early layers to overfit to suboptimal representations, creating a weak foundation for optimizing later layers (see Fig. 1.b). By contrast, Simultaneous optimization allows early layers to adapt to later ones but often suffers from large oscillations and unstable dynamics (see Fig. 1.a). This motivates us to blend the two paradigms.

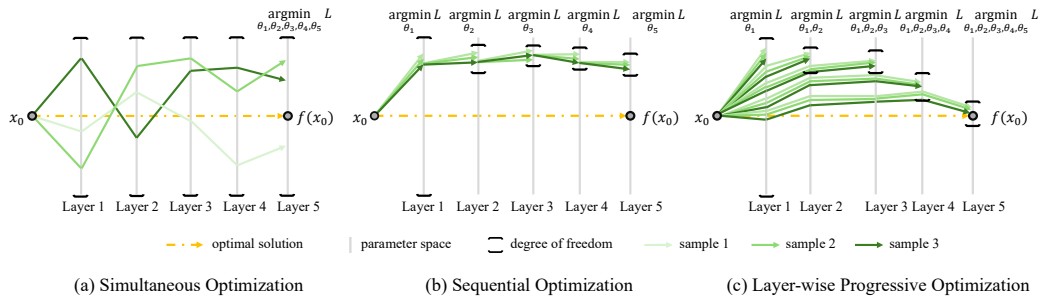

Figure 1: The optimization paths in multi-layer networks using simultaneous, sequential, and layer-wise progressive optimization Approaches.

## 4 PROGRESSIVE APPROXIMATION FOR RESIDUAL NETWORKS

### 4.1 COMBING SIMULTANEOUS AND SEQUENTIAL OPTIMIZATION

We propose Layer-wise Progressive Approximation (LPA), motivated by (Bengio et al., 2006) (You et al., 2017; Belilovsky et al., 2018; Lee et al., 2014) ,designed to implement a structured learning process where each layer's approximation capability is systematically developed while maintaining theoretical compliance with UAT requirements. More specifically, we divide a batch training into N subbatches where at the $i^{th}$ subbatch, we active the layer functions $\{G_k()\}_1^i$ while keeping the subsequent layers $\{G_k()\}_{i+1}^N$ deactivated (Alogorithm 1). It can be formulated as

$$(\theta_1, \theta_2, \cdots, \theta_N)^* = \underset{\theta_1,\theta_2,\cdots,\theta_N}{\arg\min} \underset{\cdots}{\arg\min} \underset{\theta_1,\theta_2}{\arg\min} \underset{\theta_1}{\arg\min} \mathcal{L}(\theta_1, \theta_2, \cdots, \theta_N). \quad (16)$$

---

**Algorithm 1** Layer-wise Progressive Approximation Training

1: Preprocessing layer: $G_0()$; Residual network of $L$ layers $\{G_i()\}_1^L$ and weights $\{\theta_i\}_1^L$; Linear transformation for prediction: $\mathbf{W}_o$; Training data: $\{(\mathbf{x}, \mathbf{y})\}$; Loss function: $\mathcal{L}$.
2: **for** epoch $= 1$ **to** $N_{\text{epochs}}$ **do**
3:     **for** each batch $(\mathbf{x}_b, \mathbf{y}_b)$ **do**
4:         **for** $i = 1$ **to** $L$ **do**             ▷ Layer-wise progressive approximation
5:             $\mathbf{x}_0 \leftarrow G_0(\mathbf{x}_b)$
6:             **for** $j = 1$ **to** $i$ **do**
7:                 $\mathbf{x}_j \leftarrow \mathbf{x}_{j-1} + G_j(\mathbf{x}_{j-1})$; $\mathbf{x}_i = \mathbf{x}_j$
8:             $\hat{\mathbf{y}}_i \leftarrow \mathbf{W}_o\mathbf{x}_i$                       ▷ Intermediate prediction
9:             $\mathcal{L}_i \leftarrow \mathcal{L}(\hat{\mathbf{y}}_i, \mathbf{y}_b)$
10:            Backpropagate $\mathcal{L}_i$ and update $\{\theta_1, ..., \theta_i\}$
11:     **for** each batch $(\mathbf{x}_b, \mathbf{y}_b)$ **do**
12:         $\mathbf{x}_0 \leftarrow G_0(\mathbf{x}_b)$
13:         $\mathbf{x}_L \leftarrow \text{Forward}(\mathbf{x}_0)$               ▷ Full network pass
14:         $\hat{\mathbf{y}} \leftarrow \mathbf{W}_o\mathbf{x}_L$                     ▷ Final prediction
15:         $\mathcal{L} \leftarrow \mathcal{L}(\hat{\mathbf{y}}, \mathbf{y}_b)$
16:         Backpropagate $\mathcal{L}$ and update all parameters $\{\theta_i\}_1^L$;

---

Our approach offers two key advantages. First, similar to sequential optimization, it satisfies UAT conditions by ensuring each layer $G_i()$ is properly learned before optimizing $G_{i+1}$. As formulated in Eq. (16), the layers $\{G_k()\}_1^i$ are learned in a cascaded way through the compact set transformations $K_i = \{\mathbf{x}_{i-1}+\mathcal{F}_i(\mathbf{x}_{i-1}) \mid \mathbf{x}_{i-1} \in K_{i-1}\}$. This not only aligns with the UAT conditions but also give opptunities for early layers to collaborate with the learning of the current layer $G_i()$ to adjust their learned weights. Unlike strict sequential optimization where layer learning is rigidly constrained (Fig. 1.b), this mechanism provides flexible degree of freedom that prevents entrapment in local minima (Fig. 1.c) and reduces oscillation risks inherent in simultaneous optimization (Fig. 1.a).

Second, our method establishes explicit learning objectives for intermediate layers, ensuring early coordination with the final target. Whereas simultaneous optimization only clarifies objectives at the final layer, LPA achieves provable convergence, improved accuracy, and stable training dynamics. It is worth noting that the above analysis is intended as an intuitive explanation and a comparison of training paradigms to aid a general audience. A theoretical justification of LPA's ability to improve distinguishability is provided in the Appendix F.

### 4.2 APPLICABLITY TO RESNET AND TRANSFORMER ARCHITECTURES

As long as the residual block in any RN can be written as $\mathbf{x}_i = \mathbf{x}_{i-1} + \mathbf{W}_i'\sigma(\mathbf{W}_i\mathbf{x}_{i-1} + \mathbf{b}_i) + \mathbf{b}_i'$ (Eq. (3)), the layer-wise UAT applies, making LPA directly applicable. ResNet fits this formulation naturally: neurons in layers without explicit connections can be treated as connected with zero weights, allowing convolutional blocks to be represented as FCNs (with zero-padding as needed to align layer dimensions). As Transforming Transformer architectures into this form requires much more steps, the detailed justification is provided in the Appendix B.

## 5 EXPERIMENTS

### 5.1 EXPERIMENTS ON SYNTHETIC DATASETS

We generate 6 synthetic datasets using functions as follows. There are 20,000 input-output pairs for each target function (10,000 pairs for training and 10,000 pairs for testing).

- $f(x,y) = \sin(4x) + \cos(4y)$: A multi-peak, multi-valley surface testing sensitivity to high-frequency oscillations.

- $f(x,y) = e^{-\frac{(x-0.5)^2+(y-0.5)^2}{0.1}}$: A localized Gaussian-like bump for local pattern detection.

- $f(x,y) = x^2 - y^2$: A classic hyperbolic paraboloid (saddle surface) assessing the model's ability to capture saddle-point structures.

- $f(x,y) = (|x|^{0.7} + |y|^{1.3}) \cdot \sin(4x)$: A distorted, non-uniformly oscillating surface probing the model's capability to fit non-integer powers and asymmetric combinations.

- $f(x,y) = \log(x^2+y^2+0.00001) \cdot \cos(5x)$: An oscillating surface with a central depression, testing the handling of logarithmic singularities.

- $f(x,y) = \frac{\sin(4x^2+y^2)}{(x^2+y^2+0.001)^{-2}}$: A concentric ripple pattern with maximum amplitude at the center evaluating the model's ability to fit radially decaying high-frequency oscillations.

All experiments employed a standardized 6-layer RN, where each layer comprised a 10-width MLP with ReLU activation and residual connections. We evaluated three distinct training paradigms:

- **Sequential Optimization**: The one conceptually aligned with our layer-wise analysis;

- **Simultaneous Optimization**: The conventional end-to-end training method;

- **Layer-Wise Progressive Approximation (LPA)**: Our proposed paradigm combining the strengths of the Sequental and Simultaneous approaches while mitigating their limitations.

Table 1: Performance comparison of the proposed LPA and Sequential Optimization and the coventitional end-to-end Simultaneous Optimization. The best results are in bold.

| Synthetic Datasets | High-Frequency Oscillations | | Local Pattern Detection | | Saddle-Point Structures | | Non-Int Powers Asymmetry | | Logarithmic Singularities | | Radial Decaying | |
|---|---|---|---|---|---|---|---|---|---|---|---|---|
| **Methods** | MSE ↓ | MAE ↓ | MSE ↓ | MAE ↓ | MSE ↓ | MAE ↓ | MSE ↓ | MAE ↓ | MSE ↓ | MAE ↓ | MSE ↓ | MAE ↓ |
| Sequential | 1.1354 | 0.8652 | 0.0956 | 0.2479 | 0.2662 | 0.3935 | 0.6109 | 0.6524 | 1.0023 | 0.6796 | 0.8859 | 0.8430 |
| Simultaneous | 0.7222 | 0.6928 | 0.0203 | 0.0818 | 0.0019 | 0.0353 | 0.4815 | 0.5836 | 0.8946 | 0.5680 | 0.0822 | 0.1981 |
| LPA (Ours) | **0.0487** | **0.1802** | **0.0006** | **0.0163** | **0.0007** | **0.0211** | **0.0104** | **0.0777** | **0.1095** | **0.2515** | **0.0170** | **0.0950** |

In Table 1, LPA significantly outperforms Sequential and Simultaneous by archiving 113.8x and 18.4x smaller MSE (9.7x and 3.3x smaller MAE), respectively. Figure 2 visualizes the approximated

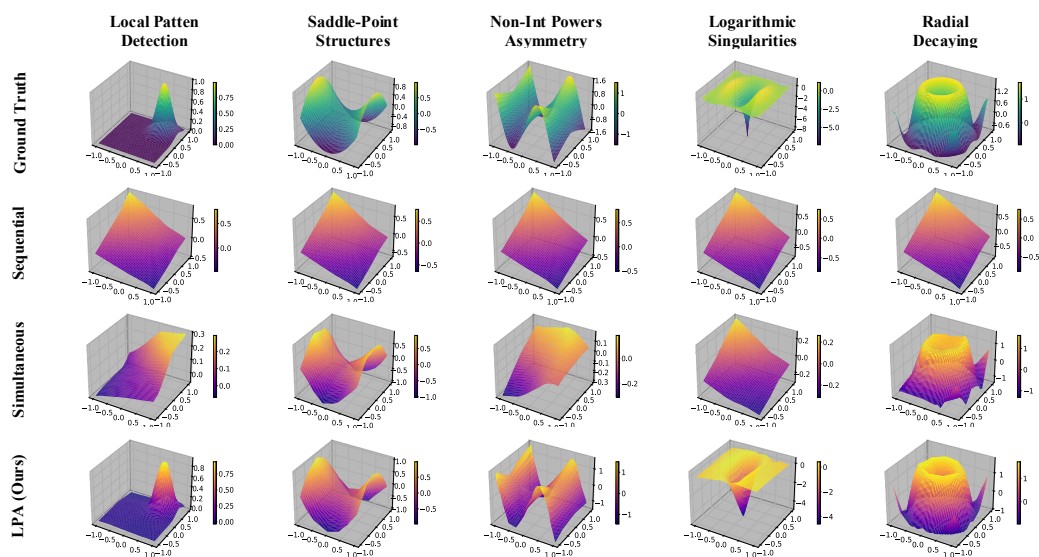

Figure 2: Comparison of approximation performance on five synthetic datasets, using surfaces generated by the target functions as ground truth.

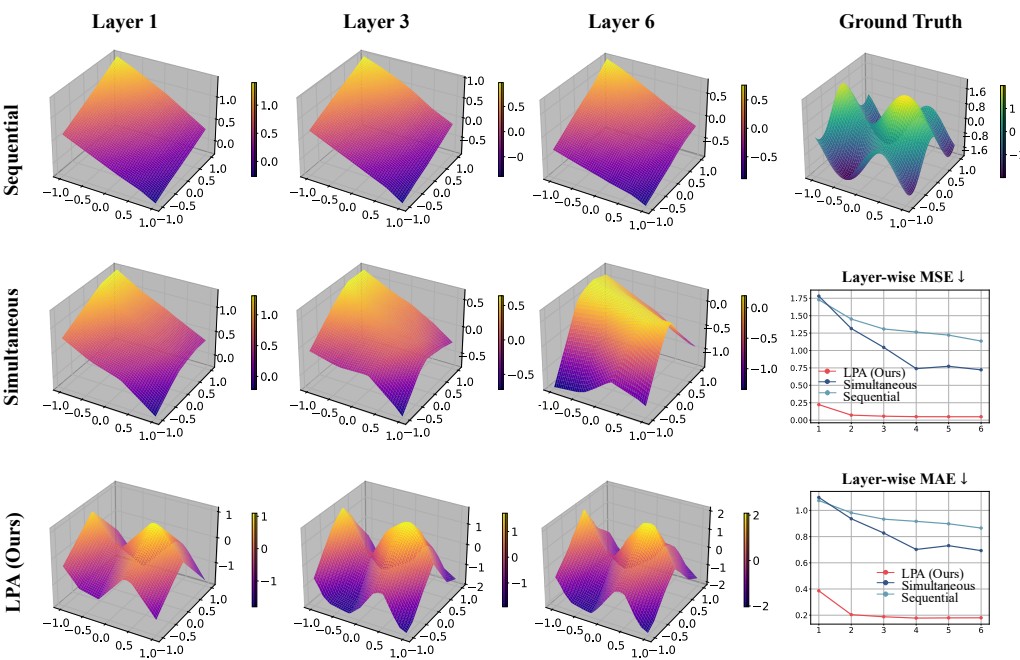

Figure 3: Layer-wise approximation results of Sequential, Simultaneous, and LPA optimizations. The target function is $f(x, y) = \sin(4x) + \cos(4y)$ with high-frequency oscillations.

surfaces under different training paradigms. LPA clearly captures the target functions far more faithfully than the other two. It validates our theoretical analysis that Sequential Optimization imposes overly restrictive constraints while Simultaneous Optimization lacks sufficient guidance, whereas LPA provides an effective balance between theoretical optimality and practical convergence.

To have a more intuitive understanding, Figure 3 presents a detailed layer-wise comparison for function $f(x, y) = \sin(4x) + \cos(4y)$. In Sequential Optimization, later layers (e.g., layers 3 and 6) were constrained by preceding layer outputs, particularly when early layers showed limited performance (e.g., Layer 1's simple slope approximation). This hierarchical constraint prevented meaningful

error correction in subsequent layers. Simultaneous Optimization afforded later layers greater flexibility but often failed to achieve a final precise approximation at the layer 6 due to insufficient coordination. In contrast, LPA successfully addressed these limitations, as evidenced by Layer 1's accurate initial approximation and Layer 6's near-perfect alignment with the target function. Quantitative metrics (layer-wise MSE and MAE) further confirmed LPA's superior convergence properties, demonstrating both faster initial progress and better final accuracy compared to alternative approaches. Additional case studies are provided in the Appendix E.

## 5.2 EXPERIMENTS ON REAL DATASETS

We conduct experiments on 3 widely used datasets: CIFAR-10, CIFAR-100, and Fashion-MNIST. We evaluate the training paradigms on two representative architectures: Residual CNNs and Transformers. For CNNs, we use the popularly adopted ResNet which consists of 18 and 34 layers with the parameters about 11.2M and 21.3M, separately. For Transformers, we use ViT models with 12 and 24 layers; each Transformer layer has 3 attention heads with a head dimension of 192, and the feed-forward network uses a hidden size of 768, approximately 5.5M parameters. All Transformer (ResNet) models are trained for 300 (50) epochs, and the results are reported in Table 4.

The proposed LPA training surpasses both Sequential and Simultaneous approaches across all datasets, yielding perfromance gains $14.26\%\pm8.33$ (up to $29.17\%$) and $3.75\%\pm2.65$ (up to $8.31\%$), respectively. We attribute this to its progressive scheme: early layers are optimized first, establishing a stronger foundation for subsequent layers and facilitating more reliable convergence.

To validate this, we examine layer-wise accuracy and loss in Figure 4. LPA reaches near-optimal performance at very early depths (around the $6^{th}$ layer on CIFAR-10 and CIFAR-100, and the $4^{th}$ layer on Fashion-MNIST), then continues to converge smoothly. In contrast, Sequential training, while stable, lacks flexibility for model-wide adjustment, and conventional Simultaneous training exhibits less smooth layer-wise progress. Notably, the simultaneous approach achieves its best results only at the final layer, leaving earlier layers under-optimized and introducing greater instability.

This is also illustrated in Figure 5, where we take layer-wise embeddings as features and apply t-SNE to assess their class separability on CIFAR-10. The early layers produced by LPA are noticeably more discriminative than those from the other two. This enables users to tailor models to their computational budgets (e.g., users can use only the first six layers and discard the rest to reduce model size without additional post-training or distillation), a valuable property for deployment.

Table 2: Performance comparison of LPA, Sequential and Simultaneous optimizations for CNN (ResNet) and Transformer (ViT) on three benchmarks of CIFAR-10 (C10), CIFAR-100 (C100), and Fashion-MNIST (FM). Best scores are in bold.

| Models | | ResNet (CNN) | | | | | | ViT (Transformers) | | | | | |
|---|---|---|---|---|---|---|---|---|---|---|---|---|---|
| Training Paradigms | | #Layers = 18 | | | #Layers = 34 | | | #Layers = 12 | | | #Layers = 24 | | |
| | | C10 | C100 | FM | C10 | C100 | FM | C10 | C100 | FM | C10 | C100 | FM |
| Sequential | Acc. ↑ | 66.36 | 41.74 | 85.01 | 85.25 | 41.44 | 85.71 | 72.85 | 48.65 | 90.00 | 73.50 | 51.18 | 90.33 |
| | Loss ↓ | 0.94 | 2.23 | 0.40 | 0.39 | 2.23 | 0.39 | 0.78 | 1.99 | 0.27 | 0.76 | 1.93 | 0.26 |
| Simultaneous | Acc. ↑ | 84.83 | 58.05 | 91.98 | 87.44 | 62.30 | 92.89 | 85.29 | 61.36 | 93.07 | 86.83 | 60.48 | 93.72 |
| | Loss ↓ | 0.44 | 1.51 | 0.22 | 0.35 | 1.33 | 0.19 | 0.46 | 1.50 | 0.19 | 0.45 | 1.74 | 0.18 |
| LPA (Ours) | Acc. ↑ | **89.71** | **65.64** | **92.54** | **91.85** | **70.61** | **93.51** | **89.37** | **64.20** | **94.25** | **90.98** | **66.08** | **94.46** |
| | Loss ↓ | 0.30 | 1.22 | 0.20 | 0.24 | 1.08 | 0.17 | 0.52 | 2.12 | 0.25 | 0.40 | 2.32 | 0.21 |

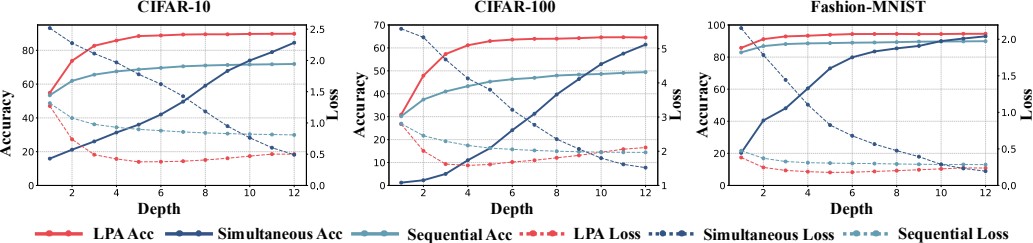

Figure 4: Comparison of layer-wise performance in accuracy and loss.

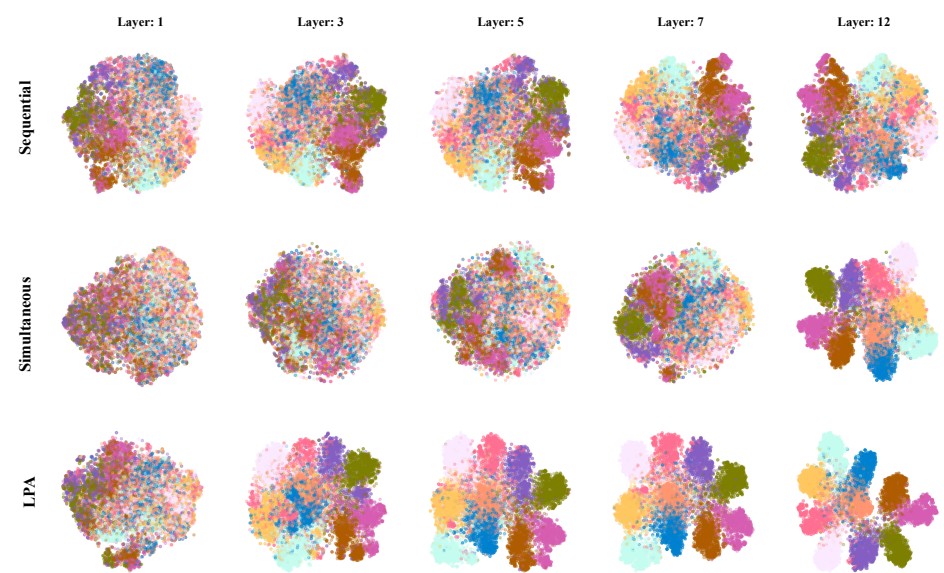

Figure 5: Comparison of class separability in layer-wise embeddings produced by Sequential, Simultaneous, and LPA optimizations.

Table 3: Comparison of performance (accuracy and loss) and model scale (number of effective layers) across Simultaneous, LPA, and Adaptive LPA for ViT (24 layers) architecture.

| Training Paradigms | Simutaneous | | | LPA | | | Adapative LPA | | |
|---|---|---|---|---|---|---|---|---|---|
| Datasets | C10 | C100 | FM | C10 | C100 | FM | C10 | C100 | FM |
| Accuracy (%) ↑ | 86.83 | 60.48 | 93.37 | 90.98 | 66.08 | 94.46 | 89.28 | 65.09 | 94.30 |
| Loss ↓ | 0.45 | 1.74 | 0.18 | 0.40 | 2.32 | 0.21 | 0.41 | 1.85 | 0.18 |
| # Effective Layers ↓ | 24 | 24 | 24 | 24 | 24 | 24 | 8 | 10 | 5 |
| Model Reduction (%) ↑ | - | - | - | - | - | - | 66.67 | 58.33 | 79.17 |

Moreover, the scalability of LPA to large-scale datasets is a crucial practical concern. While we have theoretically established the progressive approximation property of LPA and empirically validated it on CIFAR-10, CIFAR-100, and Fashion-MNIST, it remains to be verified whether this property holds in large-data regimes. To address this, we train a 12-layer ViT on ImageNet using the LPA framework. The results, summarized in Table 4, demonstrate that LPA remains effective even on ImageNet.

Table 4: Performance comparison of LPA and Simultaneous optimization for ViT on ImageNet.

| Layer | 2 | 4 | 6 | 8 | 10 | 12 |
|---|---|---|---|---|---|---|
| Sim | 0.15 | 0.33 | 0.96 | 9.55 | 48.42 | 77.75 |
| LPA | 33.19 | 56.43 | 67.66 | 73.63 | 77.34 | 78.58 |

## 5.3 IDENTIFIATION OF EFFECTIVE LAYERS USING ADATPIVE LPA

Motivated by adapting layer counts to actual accuracy gains, we ask: Is scaling truly necessary, and can the effective number of layers be discovered adaptively during training? We add a new after line 10 in Algorithm 1 as "**if** $\mathcal{L}_i < \mathcal{L}_{i-1}$ : **break**", resulting in an adaptive LPA variant that activates a new layer $(i+1)$ only if the current layer $i$ delivers a performance improvement over layer $(i-1)$. The full Adaptive LPA algorithm is provided in the Appendix D.

Table 3 reports the results. Adaptive LPA clearly outperforms conventional Simultaneous training by 2.45%, 4.61%, and 0.93% on the three benchmraks, respectively, and achieves performance

comparable to full LPA. Importantly, it automatically selects 8, 10, and 5 effective layers on the three benchmarks, respectively, enabling substantial model compression up to 79.17% from the 24-layer ViT. This suggests that simply scaling up the model is often unnecessary.

## 5.4 TRAINING, INFERENCE, AND DEPLOYMENT EFFICICENCY

The theoretical computational complexity per batch is $\mathcal{O}((1+L)L/2)$. In practice, for 24-layer ViTs, the additional layer-wise learning results in a per-batch computational overhead of $2.5\times \sim 10\times$. However, this increased cost per batch does not necessarily make LPA and Adaptive LPA more expensive overall than simultaneous training (Sim). In Figure 6, we plot test accuracy against the number of training epochs. The results demonstrate that both LPA and Adaptive LPA converge in fewer epochs than Sim. For instance, LPA (Adaptive LPA) outperform the Sim method at epochs 70 (99), 44 (50), and 125 (137), achieving test accuracies of 87.32% (87.35%), 61.21% (61.66%), and 93.91% (93.78%), respectively. This shows that training can be stopped much earlier for LPA and Adaptive LPA, offsetting the additional computational overhead.

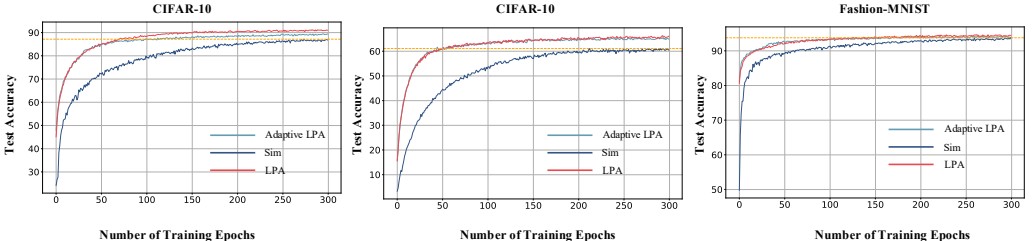

Figure 6: The test accuracy of LPA, Adaptive LPA, and Simultaneous over epochs on the CIFAR-10, CIFAR-100 and Fashion-MNIST datasets.

Additionally, it is straightforward to demonstrate that the Layer-wise UAT is recursive and remains valid when multiple layers are grouped together as a single unit (block or layer). To further investigate this, we conducted experiments applying LPA to groups of 2, 3, 4, and 6 layers, respectively. The results show that both performance and conclusions are consistent with previous findings, and the training time is comparable to that of the Sim method. For example, with a 70 training epochs and the number of layers in a LPA unit at 12 on Cifar-10 it shows a 30% of training time reduction compared to a 24-ViT of 300 epochs to reach it performance convergence. Furthermore, the accuracy has been improved to 88.37% (compared to 86.63% of the 24-ViT).

From a deployment perspective, it is worth mentioning that LPA is a "**one training run for N models**" paradigm, which makes it possible to adjust model size through pruning of later layers. These smaller, customized models achieve significantly faster inference times than the original, full-sized model. For example, our experiments demonstrate that a model compressed to one-fourth the size of the original 24-layer ViT can achieve a $4\times$ reduction in both mode size and inference time, making deployment on low-resource edge devices possible (e.g., Raspberry Pi Zero).

## 6 CONCLUSION

We extended the UAT to modern residual architectures. By expressing ResNet and Transformer blocks in a unified, FNN-compatible layer-wise form, we established that each residual layer satisfies UAT conditions on compact domains with continuous mappings. This layer-wise perspective reveals residual training as a compensatory additive process, legitimizing sequential optimization in which layers collaboratively reduce input–output discrepancy. Building on this theory, we proposed LPA, a practical training paradigm that operationalizes progressive optimization and increases the likelihood of UAT-compliant approximations at each layer. Empirically, LPA delivers consistent gains across synthetic tasks and standard benchmarks (CIFAR-10/100, Fashion-MNIST), achieving higher accuracy, faster early-layer convergence, and improved training stability relative to end-to-end baselines. Moreover, an adaptive criterion layered onto LPA automatically identifies effective depth and prunes redundant layers, enabling aggressive compression without sacrificing accuracy, challenging the assumption that larger models are inherently better.

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

## A    THE USE OF LARGE LANGUAGE MODELS

All text in this paper was drafted by the authors. GPT was used solely for error and typo checking, as well as occasional rephrasing when necessary. All content is original.

## B    UNIFING THE REPRESENTATION OF CNN AND TRANSFORMER

As we establish early in the main text, if a residual network layer can be written as Eq. (3)

$$\mathbf{x}_i = \mathbf{x}_{i-1} + \mathbf{W}_i' \sigma(\mathbf{W}_i \mathbf{x}_{i-1} + \mathbf{b}_i) + \mathbf{b}_i', \tag{17}$$

then the layer-wise UAT holds, and LPA is applicable. In this section, we detail how CNNs and Transformers can be expressed in this form.

### B.1    MATRIX AND VECOTR REPRESENTATIONS

Before diving into the details, we emphasize that in the equation, $\mathbf{W}_i'$ and $\mathbf{W}_i$ are weight matricies, while the $\mathbf{x}_i$, $\mathbf{x}_{i-1}$, $\mathbf{b}_i$ and $\mathbf{b}_i'$ are vectors. In the derivations below, all products strictly follow the standard definitions of linear algebra and matrix analysis, from a mathematical standpoint, rather than the simplified conventions often used in machine learning to describe inputs, outputs, and weights.

### B.2    CONVERSION FOR CNNS

It is evident that convolution operations can be equivalently expressed in matrix-vector form. This transformation allows us to interpret the convolution process as applying a structured, sparse matrix (derived from the convolutional kernel) to the input vector. In this formulation, each row of the matrix corresponds to a local receptive field, and the sparsity pattern encodes the weight sharing and local connectivity inherent in convolutional layers. This perspective not only unifies convolution with linear transformations but also highlights its role in hierarchically extracting spatial features from the input data. For a detailed theoretical treatment of this equivalence and its implications for representational capacity refer to Zhou (2020). Based on the matrix-vector form of convolution, it is easy to know that each layer of a multi-layer residual based CNN can be expressed as Eq. (3).

### B.3    CONVERSION FOR TRANSFORMERS

Representing Transformer operations in matrix–vector form is more involved, since they are typically expressed in matrix form (e.g., $\mathbf{Y} = \mathbf{W}\mathbf{X}$). Let us define a basic conversion $\mathbf{x} = vec(\mathbf{X})$ and $\mathbf{y} = vec(\mathbf{Y})$, where $vec(\cdot)$ denotes vectorization.

$$\mathbf{X} = \begin{vmatrix} x_{11} & x_{12} & \cdots & x_{1n} \\ x_{21} & x_{22} & \cdots & x_{2n} \\ \vdots & \vdots & \vdots & \vdots \\ x_{m1} & x_{m2} & \cdots & x_{mn} \end{vmatrix}_{m \times n} \Rightarrow \mathbf{x} = \begin{vmatrix} x_{11} \\ x_{21} \\ \vdots \\ x_{m1} \\ x_{12} \\ x_{22} \\ \vdots \\ x_{m2} \\ \vdots \\ x_{1n} \\ x_{2n} \\ \vdots \\ x_{mn} \end{vmatrix}_{mn \times 1}.$$

However, as we will soon see, this alone does not suffice to translate products like $\mathbf{Y} = \mathbf{W}\mathbf{X}$, $\mathbf{Y} = \mathbf{X}\mathbf{W}$, or $\mathbf{Y} = \mathbf{W}\mathbf{X}\mathbf{W}$ into equivalent matrix–vector expressions. We first need additional, more fundamental conversion tools.

### B.3.1 FUNDEMENTAL CONVERSION TOOLS

**Theorem B.1** (**Right Multiplication Conversion to Matrix-Vector Form**). *With* $\mathbf{X} \in \mathbb{R}^{m \times n}, \mathbf{Y} \in \mathbb{R}^{m \times p}, \mathbf{W} \in \mathbb{R}^{n \times p}$, *the right matrix multiplicaion* $\mathbf{Y} = \mathbf{X}\mathbf{W}$ *can be converted into the form of* $\mathbf{y} = \mathbf{W}'\mathbf{x}$ *where* $\mathbf{W}' = \mathbf{W}^{\top} \otimes \mathbf{I}_m$ *with* $\mathbf{I}_m \in \mathbb{R}^{m \times m}$ *representing a unit matrix.*

*Proof.* Starting with

$$\mathbf{Y}_{m \times p} = \mathbf{X}_{m \times n}\mathbf{W}_{n \times p}, \tag{18}$$

we can write

$$\begin{aligned}
vec(\mathbf{Y}_{m \times p}) &= vec(\mathbf{X}_{m \times n}\mathbf{W}_{n \times p}) \\
&= vec(\mathbf{I}_{m \times m}\mathbf{X}_{m \times n}\mathbf{W}_{n \times p}) \\
&= (\mathbf{W}^{\top} \otimes \mathbf{I}_m)vec(\mathbf{X}) \tag{19} \\
&= (\mathbf{W}^{\top} \otimes \mathbf{I}_m)\mathbf{x} \\
&= \mathbf{W}'\mathbf{x} \tag{20}
\end{aligned}$$

where $\otimes$ is Kronecker product, and $\mathbf{W}' = \mathbf{W}^{\top} \otimes \mathbf{I}_m$. In Eq. (19), Vectorization Property, which states $vec(\mathbf{A}\mathbf{B}\mathbf{C}) = (\mathbf{C}^{\top} \otimes \mathbf{A})vec(\mathbf{B})$ is used. $\square$

In a more general case, we often have a bias term as $\mathbf{Y} = \mathbf{X}\mathbf{W} + \mathbf{B}_{1 \times m}$. It is easy to see that its vertor form is:

$$\mathbf{y} = \mathbf{W}'\mathbf{x} + \mathbf{b},$$

where $\mathbf{b} = \mathbf{B} \otimes \mathbf{1}_p$ and $\mathbf{1}_m \in \mathbb{R}^{m \times 1}$ is a column vector of all ones.

**Theorem B.2** (**Left Multiplication Conversion to Matrix-Vector Form**). *With* $\mathbf{X} \in \mathbb{R}^{m \times n}, \mathbf{Y} \in \mathbb{R}^{p \times n}, \mathbf{W} \in \mathbb{R}^{p \times m}$, *the left matrix multiplicaion* $\mathbf{Y} = \mathbf{W}\mathbf{X}$ *can be converted into the form of* $\mathbf{y} = \mathbf{W}'\mathbf{x}$ *where* $\mathbf{W}' = \mathbf{I}_n \otimes \mathbf{W}$ *with* $\mathbf{I}_n \in \mathbb{R}^{n \times n}$ *representing a unit matrix.*

*Proof.* Starting with

$$\mathbf{Y}_{p \times n} = \mathbf{W}_{p \times m}\mathbf{X}_{m \times n}, \tag{21}$$

we can write

$$\begin{aligned}
vec(\mathbf{Y}_{p \times n}) &= vec(\mathbf{W}_{p \times m}\mathbf{X}_{m \times n}) \\
&= vec(\mathbf{W}_{p \times m}\mathbf{X}_{m \times n}\mathbf{I}_n) \\
&= (\mathbf{I}_n \otimes \mathbf{W}_{p \times m})vec(\mathbf{X}) \tag{22} \\
&= (\mathbf{I}_n \otimes \mathbf{W}_{p \times m})\mathbf{x} \\
&= \mathbf{W}'\mathbf{x} \tag{23}
\end{aligned}$$

where $\mathbf{W}' = \mathbf{I}_n \otimes \mathbf{W}$. In Eq. (22), Vectorization Property is used. $\square$

**Theorem B.3** (**Bi-Directional Multiplication Conversion to Matrix-Vector Form**). *With* $\mathbf{X} \in \mathbb{R}^{n \times p}, \mathbf{Y} \in \mathbb{R}^{m \times q}, \mathbf{W}_L \in \mathbb{R}^{m \times n}, \mathbf{W}_L \in \mathbb{R}^{p \times q}$, *the matrix multiplicaion* $\mathbf{Y} = \mathbf{W}_L\mathbf{X}\mathbf{W}_R$ *can be converted into the form of* $\mathbf{y} = \mathbf{W}'\mathbf{x}$ *where* $\mathbf{W}' = \mathbf{W}_R^{\top} \otimes \mathbf{W}_L$.

*Proof.* Starting with

$$\mathbf{Y} = \mathbf{W}_L\mathbf{X}\mathbf{W}_R, \tag{24}$$

where $\mathbf{W}_L \in \mathbb{R}^{m \times n}, \mathbf{X} \in \mathbb{R}^{n \times p}, \mathbf{W}_R \in \mathbb{R}^{p \times q}$, and $\mathbf{Y} \in \mathbb{R}^{m \times q}$. We proceed in two steps:

- Let $\mathbf{Z} = \mathbf{W}_L\mathbf{X}$, then $vec(\mathbf{Z}) = \mathbf{z} = (\mathbf{I}_p \otimes \mathbf{W}_L)\mathbf{x}$.

- Then, $\mathbf{Y} = \mathbf{Z}\mathbf{W}_R$, so $vec(\mathbf{Y}) = \mathbf{y} = (\mathbf{W}_R^{\top} \otimes \mathbf{I}_m)\mathbf{z}$.

Substituting the expression for $\mathbf{z}$, we get:

$$\mathbf{y} = (\mathbf{W}_R^{T} \otimes \mathbf{I}_m)(\mathbf{I}_p \otimes \mathbf{W}_L)\mathbf{x}.$$

Using the mixed-product property of the Kronecker product:
$$(\mathbf{A} \otimes \mathbf{B})(\mathbf{C} \otimes \mathbf{D}) = (\mathbf{AC}) \otimes (\mathbf{BD}),$$
we simplify:
$$(\mathbf{W}_R^\top \otimes \mathbf{I}_m)(\mathbf{I}_p \otimes \mathbf{W}_L) = \mathbf{W}_R^\top \otimes \mathbf{W}_L.$$

Thus, the final expression becomes:
$$\mathbf{y} = (\mathbf{W}_R^\top \otimes \mathbf{W}_L)\mathbf{x}.$$

$\square$

### B.3.2 FORMULATION FOR TRANSFORMERS

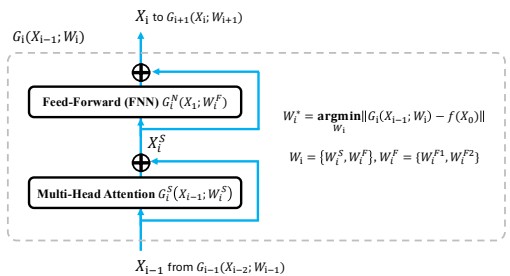

Figure 7: The $i^{th}$ layer of a Transformer illustrated using the notations defined in this paper.

Figure 7 depicts the $i^{th}$ Transformer layer, consisting of a Multi-Head Attention (MHA) block followed by a Feed-Forward (FFN) block. We now cast this process into matrix–vector form.

**Multi-Head Attention Formulation** The input $\mathbf{X}_{i-1}$ is a matrix with each column representing an input token embedding, which will be transformed in to query, key, value matrices, respectively as at the $j^{th}$ self-attention head as

$$\mathbf{Q}_{i,j} = \mathbf{W}_{i,j}^Q \mathbf{X}_{i-1},$$
$$\mathbf{K}_{i,j} = \mathbf{W}_{i,j}^K \mathbf{X}_{i-1},$$
$$\mathbf{V}_{i,j} = \mathbf{W}_{i,j}^V \mathbf{X}_{i-1},$$

where $\mathbf{X}_{i-1} \in \mathbb{R}^{d \times I}$, $\mathbf{W}_{i,j}^Q, \mathbf{W}_{i,j}^K, \mathbf{W}_{i,j}^V \in \mathbb{R}^{dh \times d}$, and $h$ is the total number of heads. It is then processed as

$$G_{i,j}^S(\mathbf{X}_{i-1}) = \mathbf{V}_{i,j} softmax\left(\frac{\mathbf{Q}_{i,j}^T \mathbf{K}_{i,j}}{\sqrt{d}}\right) \tag{25}$$

Letting $\mathbf{H}_{i,j} = softmax\left(\frac{\mathbf{Q}_{i,j}^T \mathbf{K}_{i,j}}{\sqrt{d}}\right)$, we have

$$G_{i,j}^S(\mathbf{X}_{i-1}) = \mathbf{W}_{i,j}^V \mathbf{X}_i \mathbf{H}_{i,j}. \tag{26}$$

Using **Theorem B.3**, We can the $j$-th self-attention in $i$-th layer can be written as:
$$vec(G_{i,j}^S(\mathbf{X}_{i-1})) = g_{i,j}^S(\mathbf{x}_{i-1}) = \left(\mathbf{H}_{i,j}^\top \otimes \mathbf{W}_{i,j}^V\right) \mathbf{x}_{i-1} \tag{27}$$

The whole MHA computation with all $h$ heads merged is then written
$$G_i^S(\mathbf{X}_{i-1}) = \mathbf{W}_i^O \text{Concat}[G_{i,1}^S(\mathbf{X}_{i-1})^\top, \cdots G_{i,h}^S(\mathbf{X}_{i-1})^\top]^\top + \mathbf{X}_{i-1}. \tag{28}$$

where $\mathbf{W}_i^O \in \mathbb{R}^{d \times d}$ is the weight matrix for head merger. This a right matrix multiplication, by applying **Theorem B.1**, it can be exprssed in vectorized form as

$$g_i^S(\mathbf{x}_{i-1}) = (\mathbf{I}_I \otimes \mathbf{W}_{i,O}) \begin{bmatrix} \left(\mathbf{H}_{i,1}^\top \otimes \mathbf{W}_{i,1}^V\right) \mathbf{x}_{i-1} \\ \left(\mathbf{H}_{i,2}^\top \otimes \mathbf{W}_{i,2}^V\right) \mathbf{x}_{i-1} \\ \vdots \\ \left(\mathbf{H}_{i,h}^\top \otimes \mathbf{W}_{i,h}^V\right) \mathbf{x}_{i-1} \end{bmatrix} + \mathbf{x}_{i-1}. \tag{29}$$

Note that we have replaced $G^S()$ with $g^S()$ after vectorization, as this adheres more strictly to the mathematical convention. Specifically, $G^S()$ operates on the matrix-form input $\mathbf{X}_{i-1}$, while $g^S()$ operates on the vector-form input $\mathbf{x}_{i-1}$.

Eq. (29) can be simplified as:

$$vec(G_i^S(\mathbf{X}_{i-1})) = g_i^S(\mathbf{x}_{i-1}) = (\mathbf{I}_I \otimes \mathbf{W}_{i,O}) \begin{bmatrix} \left(\mathbf{H}_{i,1}^\top \otimes \mathbf{W}_{i,1}^V\right) \\ \left(\mathbf{H}_{i,2}^\top \otimes \mathbf{W}_{i,2}^V\right) \\ \vdots \\ \left(\mathbf{H}_{i,h}^\top \otimes \mathbf{W}_{i,h}^V\right) \end{bmatrix} \mathbf{x}_{i-1} + \mathbf{x}_{i-1}. \tag{30}$$

Let

$$\mathbf{W}_i^S = (\mathbf{I}_I \otimes \mathbf{W}_{i,O}) \begin{bmatrix} \left(\mathbf{H}_{i,1}^\top \otimes \mathbf{W}_{i,1}^V\right) \\ \left(\mathbf{H}_{i,2}^\top \otimes \mathbf{W}_{i,2}^V\right) \\ \vdots \\ \left(\mathbf{H}_{i,h}^\top \otimes \mathbf{W}_{i,h}^V\right) \end{bmatrix},$$

the whole process of MHA in the $i$-th layer can be written into vector form as

$$g_i^S(\mathbf{x}_{i-1}) = \mathbf{W}_i^S \mathbf{x}_{i-1} + \mathbf{x}_{i-1}. \tag{31}$$

**Feed-Forward Formulation** The FNN component is essentially a FCN which takes input $\mathbf{X}_i^S = G^S(\mathbf{X}_{i-1})$ (or in vector form $\mathbf{x}_i^S = g^S(\mathbf{x}_{i-1})$) from the MHA component, it can be written as

$$\mathbf{X}_i = G_i^F(\mathbf{X}_i^S; \mathbf{W}_i^F) = \mathbf{W}_i^{F2} \sigma(\mathbf{W}_i^{F1} \mathbf{X}_i^S + \mathbf{b}_i^{F1}) + \mathbf{b}_i^{F2} + \mathbf{X}_i^S, \tag{32}$$

where $\mathbf{W}_i^F = \{\mathbf{W}_i^{F2}, \mathbf{W}_i^{F1}\}$ is the paramter implementation, and $\mathbf{b}_i^{F1}$ and $\mathbf{b}_i^{F2}$ is the bais for the FNN block.

With **Theorem B.1**, we can transform each part as

$$vec(\mathbf{W}_i^{F1} \mathbf{X}_i^S) = (\mathbf{I}_I \otimes \mathbf{W}_i^{F1})\mathbf{x}_i^S = \mathbf{W}_i^{F1'} \mathbf{x}_i^S,$$

$$\mathbf{b}_i^{F1'} = \mathbf{b}_i^{F1} \otimes \mathbf{1}_I,$$

$$\mathbf{b}_i^{F2'} = \mathbf{b}_i^{F2} \otimes \mathbf{1}_I,$$

$$vec(\mathbf{W}_i^{F2} \sigma(\cdot)) = (\mathbf{I}_I \otimes \mathbf{W}_i^{F2})vec(\sigma(\cdot)) = \mathbf{W}_i^{F2'} vec(\sigma(\cdot)).$$

Eq. (32) is then converted into its vector form as

$$\begin{aligned} \mathbf{x}_i = vec(G_i^F(\mathbf{X}_i^S; \mathbf{W}_i^F)) &= g_i^F(\mathbf{x}_i^S; \mathbf{W}_i^F) \\ &= \mathbf{W}_i^{F2'} \sigma(\mathbf{W}_i^{F1'} \mathbf{x}_i^S + \mathbf{b}_i^{F1'}) + \mathbf{b}_i^{F2'} + \mathbf{x}_i^S. \end{aligned} \tag{33}$$

Now we see that the entire procedure matches the form of Eq. (3), with the only difference that $\mathbf{x}_i^S$ serves as the input in place of $\mathbf{x}_{i-1}$. In fact, $\mathbf{x}_i^S$ can be viewed as a transformed version of $\mathbf{x}_{i-1}$. This is justified because, although the MHA block is central to a Transformer layer's performance, computationally it acts as a transformation of the original input $\mathbf{x}_{i-1}$ via the weight matrix $\mathbf{W}_i^S$. Put differently, Transformers form a special class of residual networks: each layer consumes a transformed version of the previous layer's output, rather than the raw output itself as in standard residual networks. Consequently, the layer-wise UAT and LPA continue to apply to Transformers. To ensure greater rigor, we have also conducted a thorough proof of how this difference affects the UAT property.

## C  A DETAILED EXAMINATION OF THE UNIVERSAL APPROXIMATION PROPERTY IN TRANSFORMERS

In this section, we analyze the universal approximation property of Transformers from a mathematical standpoint.

Based on the layer-wise UAT framework in Section 3.2, it suffices to show that a single-layer Transformer has universal approximation capability; the result then extends to deeper models. More specifically, UAT requires three conditions:

- **Compact Input**: the input domain is compact;
- **FFN-centered Network**: the model has the form of Eq. (1) with an FFN as its core;
- **Target Validity**: the target function is well-defined on a compact set.

We begin by examining the structure of a single Transformer layer and, accordingly, delineating the proof objective.

## C.1 TRANSFORMER STURCTURE AND PROOF OBJECTIVE

A single-layer Transformer consists primarily of two components: MHA and a FFN, with the overall operation expressed as $(g_i^F \circ g_i^S)(\mathbf{x}_{i-1})$, where: $g_i^S : \mathbb{R}^{dI} \to \mathbb{R}^{dI}$ denotes the feature transformation applied by the MHA module; $g_i^F : \mathbb{R}^{dI} \to \mathbb{R}^{dI}$ represents the non-linear mapping of the FFN; $\mathbf{x}_{i-1} \in K_{i-1}^I \subset \mathbb{R}^{dI}$ is the input at layer $i$, with $K_{i-1}^I = (K_{i-1})^I$ being the product space of $I$ tokens, each belonging to a compact set $K_{i-1} \subset \mathbb{R}^d$. Since finite products of compact sets are compact, $K_{i-1}^I$ is compact in $\mathbb{R}^{dI}$.

Notably, the mathematical form of the FFN component, $g_i^F$ (see Eq. (33)), aligns with the framework of the UAT (see Eq. (3)), fulfilling the **FFN-centered Network** condition.

Let the target function $f_i : K_{i-1}^I \to \mathbb{R}^{dI}$ be continuous. The key difficulty in establishing its validity is that, before the FFN $g_i^F$ receives its input $\mathbf{x}_i^S$ the MHA block $g_i^S$ applies an additional transformation, and thus the FFN does not act directly on $\mathbf{x}_{i-1}$.

Hence, $f_i$ is valid if there exists a continuous function $f_i^F$ defined on a compact domain such that

$$f_i(\mathbf{x}_{i-1}) = (f_i^F \circ g_i^S)(\mathbf{x}_{i-1}),$$

and $f_i^F$ can be approximated by $g_i^F$. Equivalently, $f_i^F$ serves as the transformed target function for the FFN module. We can now state our proof goals:

1. **Compact Input**: $K_i^S := g_i^S(K_{i-1}^I)$ is a compact subset of $\mathbb{R}^{dI}$;
2. **Target Validty**: There exists a continuous function $f_i^F : K_i^S \to \mathbb{R}^{dI}$ such that $f_i = f_i^F \circ g_i^S$.

## C.2 $K_i^S$ IS COMPACT

*Proof.* Since $K_{i-1}^I$ is a finite product of compact sets, it is compact in $\mathbb{R}^{dI}$. The transformation $g_i^S$ is composed of matrix multiplication, Kronecker product, Softmax, and vector addition. The Softmax function is continuous on $\mathbb{R}^{I \times I}$, and all linear operations are continuous; compositions and combinations of continuous functions remain continuous. Hence, $g_i^S$ is a continuous function of the input $\mathbf{x}_{i-1}$. As the continuous image of a compact set is compact, the set

$$K_i^S := g_i^S(K_{i-1}^I) = \left\{ g_i^S(\mathbf{x}_{i-1}) \mid \mathbf{x}_{i-1} \in K_{i-1}^I \right\}$$

is compact in $\mathbb{R}^{dI}$. □

## C.3 $f_i^F$ IS VALID

### C.3.1 $f_i$ IS DECOMPOSABLE

The first requirement for $f_i^F$ to be well-defined is that $f_i$ be decomposable. By the Factorization Continuity Theorem (Theorem G.1), this holds provided the following condition is satisfied:

$$g_i^S(\mathbf{x}_i^{(1)}) = g_i^S(\mathbf{x}_i^{(2)}) \implies f_i(\mathbf{x}_i^{(1)}) = f_i(\mathbf{x}_i^{(2)}), \quad \forall \mathbf{x}_i^{(1)}, \mathbf{x}_i^{(2)} \in K_{i-1}^I. \tag{34}$$

We regard this condition as satisfied because it is a learnable property reinforced by optimization. The rationale is as follows. If $g_i^S$ maps two inputs $\mathbf{x}_i^{(1)}, \mathbf{x}_i^{(2)}$ with $f_i(\mathbf{x}_i^{(1)}) \neq f_i(\mathbf{x}_i^{(2)})$ to the same representation $g_i^S(\mathbf{x}_i^{(1)}) = g_i^S(\mathbf{x}_i^{(2)})$, then the FFN (being a deterministic function) cannot simultaneously output both $f_i(\mathbf{x}_i^{(1)})$ and $f_i(\mathbf{x}_i^{(2)})$, resulting in irreducible approximation error and high loss. Consequently, gradient descent penalizes such "representation collapse," encouraging $g_i^S$ to preserve distinctions between inputs with different target values. In other words, the attention mechanism, combined with training, promotes the emergence of consistancy representations for $f_i$, ensuring that inputs differing under $g_i^S$ are not collapsed into the same latent point.

### C.3.2   EXISTENCE, CONTINUITY, AND APPROXIMABILITY OF $f_i^F$

*Proof.* Under condition in Eq. (34), we define $f_i^F : K_i^S \to \mathbb{R}^{dI}$ by

$$f_i^F(\mathbf{z}) := f_i(\mathbf{x}), \quad \text{for any } \mathbf{x} \in K_{i-1}^I \text{ such that } g_i^S(\mathbf{x}) = \mathbf{z}.$$

Condition in Eq. (34) ensures that $f_i^F$ is well-defined. By the Factorization Continuity Theorem G.1, $f_i^F$ is continuous on the compact set $K_i^S$.

Then, by the UAT, there exists a FFN $g_i^F$ that uniformly approximates $f_i^F$ to arbitrary precision.

$\square$

In summary, the MHA module $g_i^S$ continuously maps the input space $K_{i-1}^I$ to a latent representation space $K_i^S$, while the training dynamics encourage $g_i^S$ to maintain consistancy representation, ensuring that $f_i$ can be factored as $f_i^F \circ g_i^S$. The FFN $g_i^F$ then approximates $f_i^F$, completing the approximation. Therefore, a single-layer Transformer is a universal approximator.

## D   ADAPTIVE LPA

In this section, we present the algorithmic implementation of Adaptive LPA, as mentioned in Section 5.3 of the main text. The overall algorithm consists of two main stages: first, a warm-up phase, in which the entire network is trained for a small number of epochs using the standard LPA algorithm; second, the Adaptive LPA training phase. In Adaptive LPA, the decision to add a new layer is made dynamically based on whether the current layer contributes to performance improvement: if performance improves, the next layer is added; otherwise, training for the current batch is terminated early. The complete algorithm is summarized in Algorithm 2.

## E   VISUALIZATION OF PROGRESSIVE APPROXIMATION PROCESS

This section provides a visualization of the progressive approximation process for the function $f(x, y) = \frac{\sin(4x^2 + y^2)}{(x^2 + y^2 + 0.001)^{-2}}$, as discussed in Section 5.1 of the main text. As shown in Figure 8, compared to sequential and simultaneous training methods, the LPA approach achieves approximation of the target function at shallower depths. This observation is further confirmed by the MSE and MAE curves.

## F   REPRESENTATION CONSISTENCY AND OPTIMIZATION DYNAMICS IN DEEP RESIDUAL NETWORKS

A multi-layer RN can be viewed as an approximation to a target function $f : \mathbb{R}^n \to \mathbb{R}^m$ with a series of single-layer RNs $\hat{f}_N, \hat{f}_{N-1}, \cdots, \hat{f}_1$:

$$\hat{f} = \hat{f}_N \circ \hat{f}_{N-1} \circ \cdots \circ \hat{f}_1,$$

During the forward pass, an intermediate module $\hat{f}_i$ computes a representation of the input and hands it off to subsequent layers $\hat{f}_{i+1}, ..., \hat{f}_N$ for further processing. These intermediate representations

are crucial for accurately approximating the target function $f$. A key requirement is that $\hat{f}_i$ produce sufficiently distinguishable representations.

---

**Algorithm 2** Adapative Layer-wise Progressive Approximation Training

---

1: Preprocessing layer: $G_0()$; Residual network of $L$ layers $\{G_i()\}_1^L$ and weights $\{\theta_i\}_1^L$; Linear transformation for prediction: $\mathbf{W}_o$; Training data: $\{(\mathbf{x}, \mathbf{y})\}$; Loss function: $\mathcal{L}$; Pretraining epochs: $N_{\text{pre}}$; Total epochs: $N_{\text{epochs}}$; Computing accuracy: $f_{\text{acc}}$; Best accuracy: $\alpha$; Depth record for each batch: $\mathcal{D}$;
2: **for** epoch $= 1$ **to** $N_{\text{pre}}$ **do**  ▷ Warm up
3:     **for** each batch $(\mathbf{x}_b, \mathbf{y}_b)$ **do**
4:         **for** $i = 1$ **to** $L$ **do**  ▷ Layer-wise progressive approximation
5:             $\mathbf{x}_0 \leftarrow G_0(\mathbf{x}_b)$
6:             **for** $j = 1$ **to** $i$ **do**
7:                 $\mathbf{x}_j \leftarrow \mathbf{x}_{j-1} + G_j(\mathbf{x}_{j-1})$; $\mathbf{x}_i = \mathbf{x}_j$
8:             $\hat{\mathbf{y}}_i \leftarrow \mathbf{W}_o\mathbf{x}_i$  ▷ Intermediate prediction
9:             $\mathcal{L}_i \leftarrow \mathcal{L}(\hat{\mathbf{y}}_i, \mathbf{y}_b)$
10:            Backpropagate $\mathcal{L}_i$ and update $\{\theta_1, ..., \theta_i\}$
11:     **for** each batch $(\mathbf{x}_b, \mathbf{y}_b)$ **do**
12:         $\mathbf{x}_0 \leftarrow G_0(\mathbf{x}_b)$
13:         $\mathbf{x}_L \leftarrow \text{Forward}(\mathbf{x}_0)$  ▷ Full network pass
14:         $\hat{\mathbf{y}} \leftarrow \mathbf{W}_o\mathbf{x}_L$  ▷ Final prediction
15:         $\mathcal{L} \leftarrow \mathcal{L}(\hat{\mathbf{y}}, \mathbf{y}_b)$
16:         Backpropagate $\mathcal{L}$ and update all parameters $\{\theta_i\}_1^L$;
17: **for** epoch $= 1$ **to** $N_{\text{epochs}} - N_{\text{pre}}$ **do**
18:     $\mathcal{D} \leftarrow \emptyset$
19:     **for** each batch $(\mathbf{x}_b, \mathbf{y}_b)$ **do**
20:         **for** $i = 1$ **to** $L$ **do**  ▷ Layer-wise progressive approximation
21:             $\mathbf{x}_0 \leftarrow G_0(\mathbf{x}_b)$
22:             **for** $j = 1$ **to** $i$ **do**
23:                 $\mathbf{x}_j \leftarrow \mathbf{x}_{j-1} + G_j(\mathbf{x}_{j-1})$; $\mathbf{x}_i = \mathbf{x}_j$
24:             $\hat{\mathbf{y}}_i \leftarrow \mathbf{W}_o\mathbf{x}_i$  ▷ Intermediate prediction
25:             $\mathcal{L}_i \leftarrow \mathcal{L}(\hat{\mathbf{y}}_i, \mathbf{y}_b)$
26:             Backpropagate $\mathcal{L}_i$ and update $\{\theta_1, ..., \theta_i\}$
27:             **if** $f_{\text{acc}}(\hat{\mathbf{y}}_i, \mathbf{y}_b) \leq \alpha$ **then**
28:                 $\mathcal{D} \leftarrow \mathcal{D} \cup \{i\}$; break;
29:             **else**
30:                 $\alpha = f_{\text{acc}}(\hat{\mathbf{y}}_i, \mathbf{y}_b)$
31:     depth $\leftarrow \max(\mathcal{D})$
32:     **for** each batch $(\mathbf{x}_b, \mathbf{y}_b)$ **do**
33:         $\mathbf{x}_0 \leftarrow G_0(\mathbf{x}_b)$
34:         **for** $j = 1$ **to** depth **do**
35:             $\mathbf{x}_i \leftarrow \mathbf{x}_{i-1} + G_i(\mathbf{x}_{i-1})$;
36:         $\hat{\mathbf{y}}_i \leftarrow \mathbf{W}_o\mathbf{x}_i$
37:         $\mathcal{L}_i \leftarrow \mathcal{L}(\hat{\mathbf{y}}_i, \mathbf{y}_b)$
38:         Backpropagate $\mathcal{L}_i$ and update $\{\theta_1, ..., \theta_{\text{depth}}\}$

---

## F.1 Distinguishability Preservation in Intermediate Representations

In RNs, the effectiveness of intermediate representations depends not only on hierarchical abstraction, but more critically on their ability to provide embeddings consitent to the outputs of the target function $f$. Consider an arbitrary intermediate layer $i$ ($1 \leq i \leq N$), and define the encoding path up to layer $i - 1$ as:

$$\phi_i = \hat{f}_{i-1} \circ \cdots \circ \hat{f}_1,$$

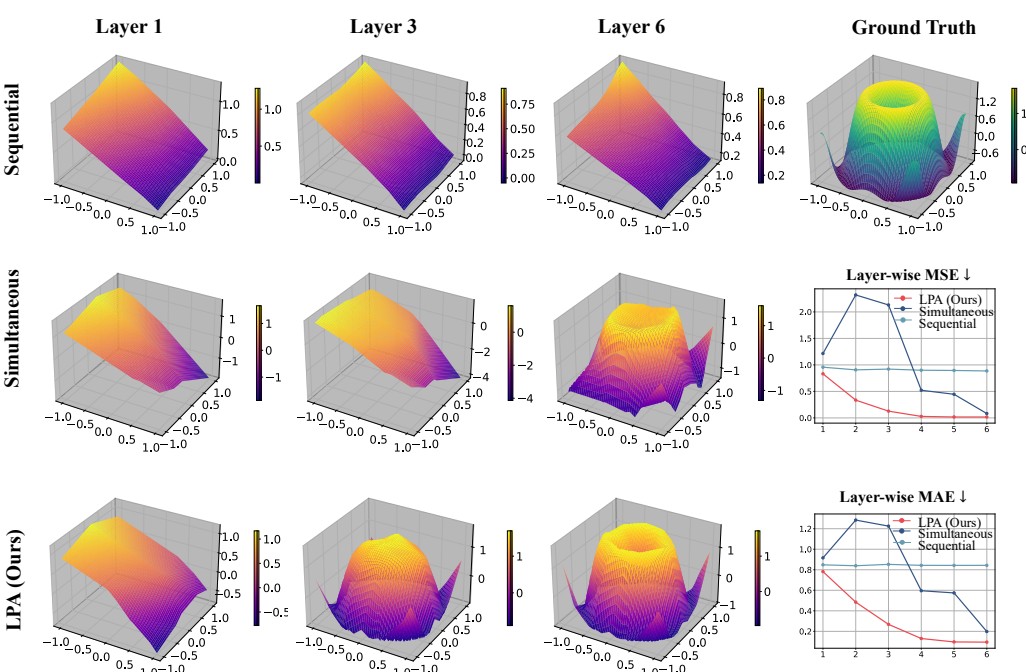

Figure 8: Layer-wise approximation results of Sequential, Simultaneous, and LPA optimizations. The target function is $f(x, y) = \frac{sin(4x^2+y^2)}{(x^2+y^2+0.001)^{-2}}$ with radial decaying.

which maps the input $\mathbf{x} \in \mathbb{R}^n$ to its representation at the output of the $i - 1$-th layer. The decoding path from layer $i$ to the output is defined as:

$$g_i = \hat{f}_N \circ \cdots \circ \hat{f}_i.$$

Thus, the network computes $\hat{f} = g_i \circ \phi_i$, where $\hat{f}$ is an approximation to the target function $f$.

Assume there are inputs $\mathbf{x}_0^{(1)}, \mathbf{x}_0^{(2)}$ with $f(\mathbf{x}_0^{(1)}) \neq f(\mathbf{x}_0^{(2)})$, yet $\phi_i(\mathbf{x}_0^{(1)}) = \phi_i(\mathbf{x}_0^{(2)})$. Then, due to the deterministic nature of $g_i$, the network will produce identical final estimates, $\hat{f}(\mathbf{x}_0^{(1)}) = \hat{f}(\mathbf{x}_0^{(2)})$, contradicting the ground truth $f(\mathbf{x}_0^{(1)}) \neq f(\mathbf{x}_0^{(2)})$. This is an indication that the optimization of the network has not converged to its optimal.

Therefore, by Factorization Continuity Theorem G.1, upon convergence (ideally $\hat{f} = f$), $\phi_i$ must satisfy:

$$\phi_i(\mathbf{x}_0^{(1)}) = \phi_i(\mathbf{x}_0^{(2)}) \implies f(\mathbf{x}_0^{(1)}) = f(\mathbf{x}_0^{(2)}), \quad \forall \mathbf{x}_0^{(1)}, \mathbf{x}_0^{(2)} \in \mathbb{R}^n. \tag{35}$$

This condition is a necessary condition for $f$ can be represented by $\hat{f}$: if violated, $f$ lies outside the expressive capacity of the $f$.

In practice, while $\hat{f}$ is typically only required to approximate $f$ rather than match it exactly, significant issues still arise when $\phi_i(\mathbf{x}_0^{(1)}) \approx \phi_i(\mathbf{x}_0^{(2)})$ but $f(\mathbf{x}_0^{(1)})$ and $f(\mathbf{x}_0^{(2)})$ differ substantially. In such cases, the decoder $g_i$ must generate vastly different outputs from nearly identical inputs, demanding extreme sensitivity to small input perturbations. This not only increases the difficulty of approximation but also risks training instability. Hence, even in approximate learning, intermediate representations should avoid mapping inputs with significantly different targets into overly similar regions of the representation space.

Equivalently, this principle can be expressed more directly by taking the conv erse-negative proposition of Eq. (35) as:

$$f(\mathbf{x}_0^{(1)}) \neq f(\mathbf{x}_0^{(2)}) \implies \phi_i(\mathbf{x}_0^{(1)}) \neq \phi_i(\mathbf{x}_0^{(2)}), \quad \forall \mathbf{x}_0^{(1)}, \mathbf{x}_0^{(2)} \in \mathbb{R}^n.$$

We refer to this necessary condition as $\hat{f}$-distinguishability, it means that the intermediate representation $\phi_i$ must preserve the distinguishability of inputs with distinct true target values under $f$. Notably, the condition is asymmetric: when $f(\mathbf{x}_0^{(1)}) = f(\mathbf{x}_0^{(2)})$, multiple inputs may share similar or even identical representations. Such over-separation is harmless and often beneficial, as it allows the network to encode auxiliary information that may aid generalization.

In summary, maintaining distinguishability between intermediate representations with distinct target is a key design principle for efficient approximation. The most critical failure mode is false merging, in which inputs with different targets are irreversibly collapsed into the same or highly similar representations. Once such structural loss occurs, no subsequent layer can recover the necessary distinctions, rendering the error uncorrectable.

### F.2 ANALYSIS UNDER SIMULTANEOUS OPTIMIZATION

In the simultaneous optimization paradigm, all layer parameters are optimized jointly to minimize the discrepancy between the final output and the target function. To illustrate the issue before convergence, consider a simplified setting with a bacth consisting of only two training examples of $\mathbf{x}_0^{(1)}$ and $\mathbf{x}_0^{(2)}$ with grounth truth $f(\mathbf{x}_0^{(1)}) \neq f(\mathbf{x}_0^{(2)})$. The loss then becomes

$$\mathcal{L} = \frac{1}{2} \sum_{j=1}^{2} \|f(\mathbf{x}_0^{(j)}) - \hat{f}(\mathbf{x}_0^{(j)})\|^2.$$

Prior to convergence, suppose the network produces the same intermediate representation for both samples, i.e., $\mathbf{x}_{i-1}^{(1)} = \mathbf{x}_{i-1}^{(2)} = \mathbf{z}$. By determinism of the network, it follows that $\hat{f}(\mathbf{x}_0^{(1)}) = \hat{f}(\mathbf{x}_0^{(2)}) =: \hat{\mathbf{y}}$. Consequently, no matter how the subsequent layers are adjusted, the network cannot simultaneously approximate the distinct targets $f(\mathbf{x}_0^{(1)})$ and $f(\mathbf{x}_0^{(2)})$ in the feed-forward process. The loss in this case is computed as:

$$\mathcal{L} = \frac{1}{2} \left( \|f(\mathbf{x}_0^{(1)}) - \hat{\mathbf{y}}\|^2 + \|f(\mathbf{x}_0^{(2)}) - \hat{\mathbf{y}}\|^2 \right).$$

which achieves its minimum when $\hat{y} = \frac{1}{2}(f(\mathbf{x}_0^{(1)}) + f(\mathbf{x}_0^{(2)}))$, yielding:

$$\min_{\{\hat{f}_j\}_i^N} \mathcal{L} = \frac{1}{4} \|f(\mathbf{x}_0^{(1)}) - f(\mathbf{x}_0^{(2)})\|^2 > 0.$$

This yields an upper bound (and an irreducible error) when optimization is restricted to updating only the parameters of layers on and after layer $i$ (i.e., $\hat{f}_i$ through $\hat{f}_N$). In other words, this error can only be corrected by updating the parameters of earlier layers ($\hat{f}_0$ through $\hat{f}_{i-1}$). In the simultaneous paradigm, however, those updates cannot occur until the loss is computed at the final layer, backpropagated through the entire network, and finally reaches the early layers. This delay can inject additional noise from later layers, potentially leading to oscillations.

### F.3 ANALYSIS UNDER LPA OPTIMIZATION

In the simultaneous optimization paradigm, representational collapse at an intermediate layer induces an irreducible approximation error, and correction in earlier layers is delayed until the final loss is backpropagated through the entire network. In contrast, the LPA actively prevents such collapse by introducing intermediate supervision, thereby enforcing $\hat{f}$-distinguishability at each layer from the moment representations are formed.

Under LPA, the objective at layer $i$ is to minimize the discrepancy between the current layer's output and the target function:

$$\mathcal{L}_i = \frac{1}{2} \sum_{j=1}^{2} \left\| f(\mathbf{x}_0^{(j)}) - \hat{f}_i(\mathbf{x}_{i-1}^{(j)}) \right\|^2,$$

where $\hat{\mathbf{y}}^{(j)} = \hat{f}_i(\mathbf{x}_0^{(j)})$ denotes the representation produced by layer $i$. Consider a mini-batch consisting of two inputs, $\mathbf{x}_0^{(1)}$ and $\mathbf{x}_0^{(2)}$, with distinct ground-truth outputs: $f(\mathbf{x}_0^{(1)}) \neq f(\mathbf{x}_0^{(2)})$. Suppose

that, during early training, the composition $\hat{f}_i \circ \cdots \circ \hat{f}_1$ fails to distinguish these inputs, resulting in identical representations at layer $i$:

$$\hat{f}_i(\mathbf{x}_{i-1}^{(1)}) = \hat{\mathbf{y}}^{(1)} = \hat{\mathbf{y}}^{(2)} = \hat{f}_i(\mathbf{x}_{i-1}^{(2)}) =: \hat{\mathbf{y}}.$$

Given that $f(\mathbf{x}_0^{(1)}) \neq f(\mathbf{x}_0^{(2)})$, this equality implies a representational collapse that layer $i$ maps distinct inputs to the same point, violating the requirement for faithful function approximation.

The corresponding layer-wise loss becomes:

$$\mathcal{L}_i = \frac{1}{2} \left( \|f(\mathbf{x}_0^{(1)}) - \hat{\mathbf{y}}\|^2 + \|f(\mathbf{x}_0^{(2)}) - \hat{\mathbf{y}}\|^2 \right),$$

which attains its minimum when $\hat{\mathbf{y}} = \frac{1}{2} \left( f(\mathbf{x}_0^{(1)}) + f(\mathbf{x}_0^{(2)}) \right)$, yielding:

$$\min_{\hat{f}_i} \mathcal{L}_i = \frac{1}{4} \|f(\mathbf{x}_0^{(1)}) - f(\mathbf{x}_0^{(2)})\|^2 > 0.$$

Crucially, unlike in simultaneous optimization, this loss $\mathcal{L}_i$ is computed at layer $i$ and immediately backpropagated through the subnetwork $\hat{f}_i \circ \cdots \circ \hat{f}_1$. The resulting gradient signal directly updates all preceding layers ($\hat{f}_1$ through $\hat{f}_i$), prompting them to refine their transformations so that $\hat{\mathbf{y}}^{(1)}$ and $\hat{\mathbf{y}}^{(2)}$ become distinguishable.

Thus, rather than waiting for errors to propagate from the global output, LPA proactively detects and corrects representational failures at their site of occurrence. This corrective mechanism ensures that $\hat{f}$-distinguishability is explicitly enforced throughout the network hierarchy, preventing persistent collapse and creating a well-conditioned learning trajectory for downstream layers. As a result, LPA not only avoids the irreducible error induced by early-layer collapse but also urge more targeted parameter updates for earlier layers.

## F.4 SUMMARY

The preceding analyses reveal a difference in how Simultaneous and LPA optimization enforce $f$-distinguishability across layers.

In Simultaneous optimization, the preservation of distinguishability is an emergent property of global error minimization. While the final loss $\mathcal{L}$ implicitly penalizes representational collapse through backpropagation, the gradient signal is:

- Delayed: Only available after full forward propagation;
- Attenuated: Subject to vanishing gradients in deep networks;
- Global: Requires coordination across all layers, making early corrections inefficient.

This results in the network can only respond to merging errors after they propagate to the output, often too late for effective correction in shallow layers.

In contrast, LPA introduces a prograssive strategy:

- At each step $i$, the intermediate loss $\mathcal{L}_i$ provides explicit supervision directly at layer $i$, enforcing $f$-distinguishability in $\mathbf{x}_i$;
- The gradient of $\mathcal{L}_i$ propagates implicitly to all preceding layers, regularizing their representations to support the current prediction;
- By progressively increasing $i$ within each batch, LPA enables cumulative refinement: shallow layers are optimized multiple times under increasingly deeper supervision, leading to robust and stable representation learning.

Crucially, LPA's immediate optimization, which updats parameters right after computing $\mathcal{L}_i$, avoids gradient interference from deeper layers and ensures that errors are corrected as soon as they become observable. This bottom-up, layer-wise learning allows the network to build a reliable representational foundation before advancing to more abstract levels.

### F.5 UNIFIED INTERPRETATION VIA OPTIMIZATION DYNAMICS: EMERGENCE OF $f$-DISTINGUISHABILITY

Under both simultaneous and LPA optimization, representation collapse induces irreducible error. Since gradient descent inherently minimizes the loss, it implicitly penalizes parameter configurations that lead to information loss. This mechanism drives the network to gradually establish $f$-consistancy in intermediate representation during training—inputs with different $f$-values become increasingly separated in the intermediate representation space.

Notably, the residual connection plays a crucial role: by preserving the identity path, it maintains access to the original input information, reducing the risk of information loss and providing structural support for the emergence of $f$-consistancy.

### F.6 GENERALITY: OPTIMIZATION-INDUCED SEPARATION

This mechanism is not limited to residual networks or specific optimization strategies. In any deep model where subsequent modules depend deterministically on earlier representations (i.e., $\mathbf{x}_j = T_j(\mathbf{x}_{j-1})$), representational collapse leads to unrecoverable errors. Thus, both end-to-end joint training and layer-wise optimization use loss feedback to drive the network away from information merging.

We term this universal phenomenon optimization-induced consistancy: the training process itself acts as an implicit regularizer, encouraging the network to preserve task-relevant information structures during representation learning. This provides a unified perspective for understanding generalization, representation evolution, and universal approximation in deep networks.

## G FACTORIZATION CONTINUITY THEOREM

**Theorem G.1 (Factorization and Continuity).** *Let* $\mathbf{I}_n \subset \mathbb{R}^n$ *be compact,* $f : \mathbf{I}_n \to \mathbb{R}^m$ *continuous, and* $T : \mathbf{I}_n \to Z$ *continuous with* $Z \subset \mathbb{R}^d$ *compact. Then:*

- *(**Existence of** $g$) There exists a function* $g : T(\mathbf{I}_n) \to \mathbb{R}^m$ *such that* $f = g \circ T$ *if and only if*

$$T(\mathbf{x}^{(1)}) = T(\mathbf{x}^{(2)}) \implies f(\mathbf{x}^{(1)}) = f(\mathbf{x}^{(2)}), \quad \forall \mathbf{x}^{(1)}, \mathbf{x}^{(2)} \in \mathbf{I}_n.$$

- *(**Continuity of** $g$) If such a* $g$ *exists, then* $g$ *is continuous on the compact set* $T(\mathbf{I}_n)$.

If the above assumption holds, then since $g$ is continuous on the compact set $T(\mathbf{I}_n) \subseteq Z$, it follows that $g$ can be approximated by a neural network. We now provide the proof.

*Proof.* Let $\mathbf{I}_n \subseteq \mathbb{R}^n$ be a nonempty compact set, $f : \mathbf{I}_n \to \mathbb{R}^m$ a continuous function, and $T : \mathbf{I}_n \to \mathbb{R}^d$ a continuous mapping. Define the image set:

$$Z := T(\mathbf{I}_n) = \{T(\mathbf{x}) \mid \mathbf{x} \in \mathbf{I}_n\} \subseteq \mathbb{R}^d.$$

Since $\mathbf{I}_n$ is compact and $T$ is continuous, $Z$ is also compact (the continuous image of a compact set is compact).

Our purpose is to prove that there exists a function $g : Z \to \mathbb{R}^m$ such that:

$$f(\mathbf{x}) = g(T(\mathbf{x})) = (g \circ T)(\mathbf{x}), \quad \forall \mathbf{x} \in \mathbf{I}_n,$$

i.e., $f = g \circ T$. $\square$

**Lemma G.2 (Existence).** *Let* $T : \mathbf{I}_n \to \mathbb{R}^d$ *be an arbitrary mapping and* $f : \mathbf{I}_n \to \mathbb{R}^m$. *If*

$$\forall \mathbf{x}^{(1)}, \mathbf{x}^{(2)} \in \mathbf{I}_n, \quad T(\mathbf{x}^{(1)}) = T(\mathbf{x}^{(2)}) \implies f(\mathbf{x}^{(1)}) = f(\mathbf{x}^{(2)}),$$

*then there exists a function* $g : T(\mathbf{I}_n) \to \mathbb{R}^m$ *such that* $f = g \circ T$.

*Proof.* For each $\mathbf{z} \in Z$, define $g(\mathbf{z}) := f(\mathbf{x})$ for any $\mathbf{x} \in \mathbf{I}_n$ with $T(\mathbf{x}) = \mathbf{z}$. Such an $\mathbf{x}$ exists because $Z = T(\mathbf{I}_n)$. This definition is well-defined: if $T(\mathbf{x}^{(1)}) = T(\mathbf{x}^{(2)}) = \mathbf{z}$, then by assumption $f(\mathbf{x}^{(1)}) = f(\mathbf{x}^{(2)})$. So $g(\mathbf{z})$ is independent of the choice of preimage.

By construction, $g(T(\mathbf{x})) = f(\mathbf{x})$ for all $\mathbf{x} \in \mathbf{I}_n$, so there exists a function $g$ fulfilling $f = g \circ T$.

**Necessity:** Suppose $f = g \circ T$, i.e., $f(\mathbf{x}) = g(T(\mathbf{x}))$. If $T(\mathbf{x}^{(1)}) = T(\mathbf{x}^{(2)})$, then:
$$f(\mathbf{x}^{(1)}) = g(T(\mathbf{x}^{(1)})) = g(T(\mathbf{x}^{(2)})) = f(\mathbf{x}^{(2)}).$$
This proves the necessity.

**Sufficiency:** Suppose that
$$T(\mathbf{x}^{(1)}) = T(\mathbf{x}^{(2)}) \implies f(\mathbf{x}^{(1)}) = f(\mathbf{x}^{(2)}), \quad \forall \mathbf{x}^{(1)}, \mathbf{x}^{(2)} \in \mathbf{I}_n.$$
We aim to construct a function $g : Z := T(\mathbf{I}_n) \to \mathbb{R}^m$ such that $f = g \circ T$. Define $g$ as follows: for each $\mathbf{z} \in Z$, choose any $\mathbf{x} \in \mathbf{I}_n$ such that $T(\mathbf{x}) = \mathbf{z}$ (such $\mathbf{x}$ exists by definition of $Z$), and set
$$g(\mathbf{z}) := f(\mathbf{x}).$$

We now show that $g$ is well-defined, i.e., the value $g(\mathbf{z})$ does not depend on the choice of preimage $\mathbf{x}$. Let $\mathbf{x}^{(1)}, \mathbf{x}^{(2)} \in \mathbf{I}_n$ be such that $T(\mathbf{x}^{(1)}) = T(\mathbf{x}^{(2)}) = \mathbf{z}$. Then by assumption, $f(\mathbf{x}^{(1)}) = f(\mathbf{x}^{(2)})$. Hence, regardless of which preimage we choose, $g(\mathbf{z}) = f(\mathbf{x})$ is the same. Therefore, $g$ is well-defined. Finally, by construction, for every $\mathbf{x} \in \mathbf{I}_n$, we have
$$g(T(\mathbf{x})) = f(\mathbf{x}),$$
since $g(T(\mathbf{x}))$ was defined precisely as $f(\mathbf{x})$ when $\mathbf{x}$ was chosen as the preimage of $T(\mathbf{x})$. Thus, $f = g \circ T$. $\qquad\square$

**Lemma G.3** (Uniqueness). *If $g_1, g_2 : Z := T(\mathbf{I}_n) \to \mathbb{R}^m$ both satisfy $f = g_1 \circ T = g_2 \circ T$, then $g_1 = g_2$ on $Z$.*

*Proof.* Assume $g_1$ and $g_2$ satisfy:
$$f(\mathbf{x}) = g_1(T(\mathbf{x})), \quad f(\mathbf{x}) = g_2(T(\mathbf{x})), \quad \forall \mathbf{x} \in \mathbf{I}_n.$$
We aim to show $g_1(\mathbf{z}) = g_2(\mathbf{z})$ for all $\mathbf{z} \in Z$. Let $\mathbf{z} \in Z$ be arbitrary. Since $Z = T(\mathbf{I}_n)$, there exists $\mathbf{x} \in \mathbf{I}_n$ such that $T(\mathbf{x}) = \mathbf{z}$. Then:
$$g_1(\mathbf{z}) = g_1(T(\mathbf{x})) = f(\mathbf{x}), g_2(\mathbf{z}) = g_2(T(\mathbf{x})) = f(\mathbf{x}).$$
Hence $g_1(\mathbf{z}) = f(\mathbf{x}) = g_2(\mathbf{z})$. Since $\mathbf{z}$ was arbitrary, $g_1 = g_2$ on $Z$. Therefore, the function $g : Z \to \mathbb{R}^m$ satisfying $f = g \circ T$ is unique. $\qquad\square$

**Lemma G.4** (Continuity). *Let $\mathbf{I}_n \subseteq \mathbb{R}^n$ be a nonempty compact set, $f : \mathbf{I}_n \to \mathbb{R}^m$ continuous, $T : \mathbf{I}_n \to \mathbb{R}^d$ continuous, and suppose:*
$$\forall \mathbf{x}^{(1)}, \mathbf{x}^{(2)} \in \mathbf{I}_n, \quad T(\mathbf{x}^{(1)}) = T(\mathbf{x}^{(2)}) \implies f(\mathbf{x}^{(1)}) = f(\mathbf{x}^{(2)}).$$
*Let $Z := T(\mathbf{I}_n) \subseteq \mathbb{R}^d$. Then there exists a unique function $g : Z \to \mathbb{R}^m$ such that $f = g \circ T$. We claim that $g$ is continuous on $Z$.*

*Proof.* Suppose, for contradiction, that $g$ is discontinuous at some point $\mathbf{z}_0 \in Z$. Then there exists $\varepsilon_0 > 0$ and a sequence $\{\mathbf{z}_k\} \subset Z$ such that $\mathbf{z}_k \to \mathbf{z}_0$, but
$$\|g(\mathbf{z}_k) - g(\mathbf{z}_0)\| \geq \varepsilon_0, \quad \forall k \in \mathbb{N}.$$
Since $\mathbf{z}_k \in Z = T(\mathbf{I}_n)$, for each $k$, there exists $\mathbf{x}_k \in \mathbf{I}_n$ such that $T(\mathbf{x}_k) = \mathbf{z}_k$. As $\mathbf{I}_n$ is compact, the sequence $\{\mathbf{x}_k\}$ has a convergent subsequence $\{\mathbf{x}_{k_j}\}$ such that
$$\mathbf{x}_{k_j} \to \mathbf{x}^* \in \mathbf{I}_n, \quad \text{as } j \to \infty.$$
By continuity of $T$, we have
$$T(\mathbf{x}_{k_j}) \to T(\mathbf{x}^*).$$
But $T(\mathbf{x}_{k_j}) = \mathbf{z}_{k_j}$, and since $\mathbf{z}_k \to \mathbf{z}_0$, the subsequence $\mathbf{z}_{k_j} \to \mathbf{z}_0$. By uniqueness of limits,
$$T(\mathbf{x}^*) = \mathbf{z}_0.$$
By continuity of $f$,
$$f(\mathbf{x}_{k_j}) \to f(\mathbf{x}^*).$$
Note that $f(\mathbf{x}_{k_j}) = g(T(\mathbf{x}_{k_j})) = g(\mathbf{z}_{k_j})$, and $f(\mathbf{x}^*) = g(T(\mathbf{x}^*)) = g(\mathbf{z}_0)$. Therefore,
$$g(\mathbf{z}_{k_j}) \to g(\mathbf{z}_0), \quad \text{as } j \to \infty.$$
Thus, there exists $J \in \mathbb{N}$ such that for all $j \geq J$, $\|g(\mathbf{z}_{k_j}) - g(\mathbf{z}_0)\| < \varepsilon_0$. However, this contradicts the earlier assumption that
$$\|g(\mathbf{z}_k) - g(\mathbf{z}_0)\| \geq \varepsilon_0, \quad \forall k,$$
which in particular implies $\|g(\mathbf{z}_{k_j}) - g(\mathbf{z}_0)\| \geq \varepsilon_0$ for all $j$. Hence, the assumption of discontinuity is false. Therefore, $g$ is continuous at $\mathbf{z}_0$. Since $\mathbf{z}_0 \in Z$ was arbitrary, $g$ is continuous on $Z$. $\qquad\square$

## H    Additional Information for Rebuttal

This section is intended solely as a visual aid for the rebuttal discussion and will be removed in the final version. Some content may be incorporated into the main body of the paper.

Figure 9 showes the epoch-test accuracy curves on CIFAR-10, CIFAR-100, and Fashion-MNIST, respectively. As observed, both LPA and Adaptive LPA surpass the Simultaneous method at epochs 70 (99), 44 (50), and 125 (137), achieving test accuracies of 87.32% (87.35%), 61.21% (61.66%), and 93.91% (93.78%), respectively (values in parentheses correspond to Adaptive LPA).

Figure 10 further evaluates the performance of the LPA method on CIFAR-10 under different learning rates.

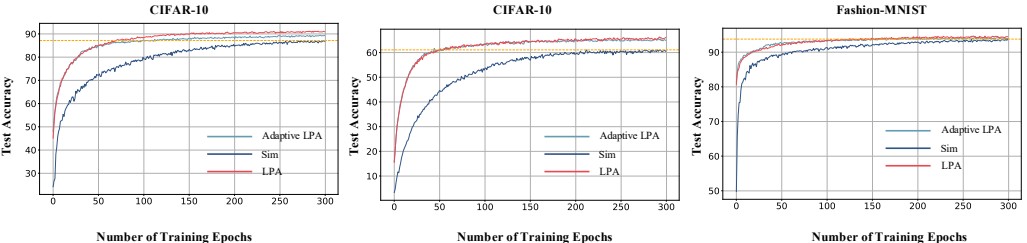

Figure 9: The test accuracy of LPA, Adaptive LPA, and Simultaneous over epochs on the CIFAR-10, CIFAR-100 and Fashion-MNIST datasets.

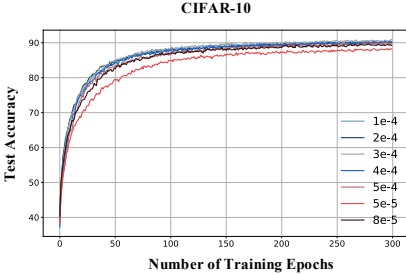

Figure 10: The affect of varying learning rates on the LPA performance in terms of test accuracy over epochs on the CIFAR-10.

## I    Proof-of-Concept Experiment on Large Language Models

To further assess LPA's generalizability to advanced residual models, we conducted a proof-of-concept experiment applying LPA to train a mini-QWen model with 1B parameters. Specifically, we extended Qwen2.5-0.5B-Instruct to a 48-layer, 1B-parameter model and trained it from scratch. Pretraining (PT) was performed on a 40GB text corpus collected from the Internet, followed by supervised fine-tuning (SFT) on a 12GB conversational dataset. Due to the significant computational demands, as the training required 55 hours on a cluster of 16 GPUs (each with 80GB memory), we limited both PT and SFT to just one epoch. Both simultaneous training and LPA were used, resulting in two separate LLMs.

Because SFT relied on conversational data, the resulting models are primarily capable of chatting in natural language, rather than instruction following or problem-solving. Thus, we evaluated the models by prompting them with MMLU questions and assessing their responses using GPT-5 as a judge, focusing on fluency, naturalness, and readability. GPT-5 rated responses on a scale from 1 to 10, according to the following criteria:

- 1–3: Poor (grammatically incorrect, incoherent, difficult to understand)
- 4–5: Below Average (some errors, awkward phrasing, but somewhat understandable)

- 6–7: Average (generally correct, may have minor issues, acceptable quality)
- 8–9: Good (fluent, natural, easy to read, with at most minor imperfections)
- 10: Excellent (perfectly fluent, natural, and readable)

We evaluated both the intermediate and final layers of each model. The results are presented in Table 5.

Table 5: Comparison of language usage in miniQWen models trained via Sim and LPA.

| Layer | 12 | 24 | 36 | 48 |
|-------|------|------|------|------|
| Sim   | 1.36 | 1.15 | 1.26 | 5.94 |
| LPA   | 4.67 | 4.61 | 4.55 | 6.43 |

The results demonstrate that the progressive property is preserved in the intermediate layers of models trained with LPA. The LPA model's conversational ability is maintained even when the model is compressed to just 12 layers. Although these findings are preliminary, they provide evidence supporting the generalizability of LPA to modern residual networks.

