# OpenReview forum: "Layer-Wise Universal Approximation and Progressive Optimization for Residual Networks"
_ICLR.cc/2026/Conference — Submitted to ICLR 2026_

### Official Review · Reviewer_yRZ7 · 2025-10-24

**Soundness:** 1
**Presentation:** 2
**Contribution:** 1
**Rating:** 2
**Confidence:** 3

**Summary:**

This paper extends the Universal Approximation Theorem (UAT), originally developed for feedforward networks, to ResNets and Transformers. The authors make three main contributions:

1. Demonstrate that layer-wise UAT can be applied to ResNets and Transformers by reformulating their block operations into matrix–vector form.

2. Propose a collaborative end-to-end and sequential training approach that leverages this theoretical connection.

3. Introduce Layer-wise Progressive Approximation (LPA), a layer-wise training method that reportedly outperforms standard end-to-end methods on synthetic datasets, CIFAR-10/100, and Fashion-MNIST.

**Strengths:**

- The related works are appropriately cited.

- The Universal Approximation Theorem is explicitly stated, providing readers with the necessary theoretical background.

- Figures 2 and 3 provide visually clear and straightforward comparisons that help illustrate the experimental results.

**Weaknesses:**

**Motivation and conceptual clarity**: The motivation is weak and conceptually confusing. The paper conflates approximation (a representational property) with optimization (a training procedure), which are distinct things. The logic connecting UAT to the proposed method, especially in the last paragraph of Section 3, is unclear and unconvincing.

**Technical soundness**: Due to the above conceptual issues, the technical grounding of the paper is questionable, making it more like a heuristic tweak. The proposed LPA algorithm essentially performs layer-wise training (Algorithm 1) given that $\mathcal{L}$ is the loss function. This has nothing to with UAT, which concerns function approximation that does not involve training.

**Presentation issues**: Figure 1 is difficult to interpret due to vague captions, and insufficient legends and labels. Experimental details on synthetic datasets are also missing.

**Limited validation**: The evaluation is restricted to small-scale datasets (CIFAR, Fashion-MNIST), without experiments on larger or more diverse tasks (e.g., ImageNet), making the practical significance unclear.

**Questions:**

1. The “gap” between prior UAT studies and the proposed approach is not clearly defined and well-motivated. The authors claim that prior works are architecture-specific (line 90). Does this mean those results were derived for modified versions of ResNet/Transformer architectures, while the authors target the vanilla versions? If so, what exactly differs, and why can’t prior results generalize?

2. What new insights are gained by stating that UAT is “feasible to RNs” at the layer level? How does this provide practical or theoretical value for understanding residual networks?

3. The terms “compensatory additive problem” and “oscillation risks” are undefined and unclear throughout the paper. Please provide formal definitions or intuitive explanations.

4. Section 3’s analysis appears trivial: it essentially restates that a multilayer fully connected network with residual connections is a universal approximator. The derivation in Eq. (15) does not seem to introduce new theoretical insight. Could the authors clarify what is genuinely novel here?

5. How does the layer-wise approximation analysis in Section 3 translate into the layer-wise training procedure in Section 4? These are conceptually different topics (representation vs. optimization).

6. The theoretical results primarily rely on fully connected networks with sigmoid activations. How do they extend to modern ResNets or Transformers with non-sigmoid activations and more complicated structures? The appendix (B.2) provides only textual justifications without explicit mathematical conversion from CNNs to FCNs. If it cannot, isn't the whole approach still architecture-specific, just falling into the limitations the authors pointed out in previous works? As such, the theoretical part does not fill the claimed gaps at all.

7. What is the computational cost or time complexity of the proposed training procedure? Since each layer requires retraining previous ones, it appears computationally heavy.

8. Prior work (e.g., Greedy Layerwise Learning Can Scale to ImageNet, ICML 2019) already demonstrated over 90% accuracy on CIFAR-10 with layer-wise training, which I believe is the sequential training in this paper. Why does your sequential optimization yield significantly lower accuracy?

---

> ### Author Response · Authors · 2025-11-25
> **Reviewer yRZ7**
>
> **Weaknesses:**
> Thank you for your feedback. We appreciate the opportunity to clarify key concepts in our work, which we believe are central to understanding its contribution. Below we address each concern with care and precision.
>
> **1. Comment:**  "Motivation and conceptual clarity: The motivation is weak and conceptually confusing. The paper conflates approximation (a representational property) with optimization (a training procedure), which are distinct things. The logic connecting UAT to the proposed method, especially in the last paragraph of Section 3, is unclear and unconvincing."
>
> **Response:** Thank you for raising this important point. We agree that approximation (a representational property) and optimization (a training process) are distinct concepts. This distinction is precisely why we have addressed them in separate sections of the paper.
>
> In Section 3, we provide a theoretical foundation demonstrating that deep networks with residual-like structures are capable of progressive function approximation at the layer level. This is a result from representation theory, grounded in universal approximation theory (UAT), and is independent of any specific training procedure. Nevertheless, this theoretical analysis clarifies that the learning objective can be reframed as learning the input-output difference (Eq. (12)), and further, that inter-layer relationships can be described as a compensatory additive system (Eq. (13)). Recognizing that both sequential and simultaneous learning paradigms are applicable to such systems, we transition in Section 4 to exploring whether an effective and equivalent training procedure can be implemented for residual networks, leading to the design of LPA.
>
> Therefore, the connection between UAT and LPA is not a conceptual conflation, but rather a deliberate link from theoretical representation to practical implementation, namely using theory to inform and guide algorithm design. We will further clarify and highlight this connection in the revised manuscript.
>
> **2. Comment:**  "Technical soundness: Due to the above conceptual issues, the technical grounding of the paper is questionable, making it more like a heuristic tweak. The proposed LPA algorithm essentially performs layer-wise training (Algorithm 1) given that $L$ is the loss function. This has nothing to with UAT, which concerns function approximation that does not involve training."
>
> **Response:** Thank you for raising this concern. As discussed above, the analysis of function approximation has enabled us to identify both the necessary conditions and the revised objectives for effective approximation. This theoretical foundation directly informed the practical design of the LPA method.
>
> **3. Comment:**  "Presentation issues: Figure 1 is difficult to interpret due to vague captions, and insufficient legends and labels. Experimental details on synthetic datasets are also missing."
>
> **Response:** Figure 1 is a conceptual illustration intended to convey the idea of progressive approximation: how each layer refines the input toward the target output. It is not meant to be a quantitative plot but rather a visual aid to support the theoretical intuition. The actual performance results are reported in Tables 1 and 2, where we show that LPA consistently outperforms baseline methods on both CIFAR-10/100 and Fashion-MNIST, confirming the validity of the depicted behavior.
> Regarding the synthetic dataset experiments:
> 	We generated 10,000 training pairs and 10,000 testing pairs for each function.
> 	The model was trained for 50 epochs with the same settings.
> We plan to release the code publicly upon acceptance, so all experimental conditions will be fully reproducible.
>
> **4. Comment:** "Limited validation: The evaluation is restricted to small-scale datasets (CIFAR, Fashion-MNIST), without experiments on larger or more diverse tasks (e.g., ImageNet), making the practical significance unclear."
>
> **Response:** Thank you for your valuable suggestion. We appreciate the significance of scalability and have conducted experiments on ImageNet with a 12-layer ViT model. The results, shown below, align with our observations on CIFAR-10/100 and Fashion-MNIST: models trained with LPA display progressive behavior, with earlier layers achieving performance similar to those trained with the simultaneous paradigm. We also plan to extend our evaluation to a 24-layer ViT and will include these updated results in the revised manuscript.
> | Layer | 2     | 4     | 6     | 8     | 10    | 12    |
> |-------|-------|-------|-------|-------|-------|-------|
> | Sim   | 0.152 | 0.336 | 0.964 | 9.558 | 48.428| 77.752|
> | LPA   | 33.196| 56.432| 67.660| 73.636| 77.344| 78.586|

---

> ### Author Response · Authors · 2025-11-25
>
> **Questions:**
>
> **1. Comment:** "The “gap” between prior UAT studies and the proposed approach is not clearly defined and well-motivated. The authors claim that prior works are architecture-specific (line 90). Does this mean those results were derived for modified versions of ResNet/Transformer architectures, while the authors target the vanilla versions? If so, what exactly differs, and why can’t prior results generalize?"
>
> **Response:** We thank you for highlighting this important issue. They typically proved universal approximation (UAT) for specific network structures, such as ResNet or Transformer variants, by constructing ad hoc approximations tailored to their architecture. The implementation way is that they write different architecture into corresponding mathematical format and then prove that format fulfilled UAT.
> In contrast, our contribution lies in establishing a general framework that unifies diverse architectures (e.g., ResNet, Transformer) under a common mathematical form: UAT format. As shown in Appendix B, both residual networks and transformers can be reformulated into a unified functional structure where each layer acts as a residual update on the input representation. This unified form shares similar format with classical fully connected networks, allowing us to extend the standard UAT result to these modern architectures.
>
> 2. **Comment:** "What new insights are gained by stating that UAT is “feasible to RNs” at the layer level? How does this provide practical or theoretical value for understanding residual networks? "
>
> **Response:** Its significance lies in providing a formal characterization of how deep networks can learn progressively, which has not been done before. Specifically, we show that:
>
> - Compensatory Additive Problem: In residual networks, each layer computes a residual and they are composed of add. The "compensatory" nature arises when earlier layers fail to capture sufficient information, forcing later layers to compensate by learning residuals.
>
> - Oscillation Risks: Refers to the instability observed during end-to-end training (simultaneous), where the loss curve exhibits fluctuations (as seen in Figure 3).
>
> 4 & 5. **Comment:**  "Section 3’s analysis appears trivial: it essentially restates that a multilayer fully connected network with residual connections is a universal approximator. The derivation in Eq. (15) does not seem to introduce new theoretical insight. Could the authors clarify what is genuinely novel here?
>
> How does the layer-wise approximation analysis in Section 3 translate into the layer-wise training procedure in Section 4? These are conceptually different topics (representation vs. optimization)."
>
> **Response:** Section 3 presents a theoretical analysis of how deep networks can perform progressive approximation. Section 4 derives the LPA algorithm as a direct implementation of this theory: each layer is trained to minimize the residual error.
> Thus, the analysis is not trivial; it provides a principled justification for layer-wise training, explaining why such methods work and how they differ from end-to-end optimization. It connects representation (what the network learns) with optimization (how it learns), which is precisely the conceptual gap we aim to bridge.
>
> 6. **Comment:** "The theoretical results primarily rely on fully connected networks with sigmoid activations. How do they extend to modern ResNets or Transformers with non-sigmoid activations and more complicated structures? The appendix (B.2) provides only textual justifications without explicit mathematical conversion from CNNs to FCNs. If it cannot, isn't the whole approach still architecture-specific, just falling into the limitations the authors pointed out in previous works? As such, the theoretical part does not fill the claimed gaps at all."
>
> **Response:** We thank you for raising this important technical concern. We did not rely on fully connected networks and we have justified in the Appendix that the results hold for Transformers.
> With respect to sigmoid activations, the conclusion on original UAT is not limited to sigmoid function. As long as conditions  required by UAT are fulfilled, any non-sigmoid function can be used.
> We have provided the formal transformation method from CNN to FCN in paper and you can refer to the work "Universality of deep convolutional neural networks" (Applied and Computational Harmonic Analysis, 2020).

---

> ### Author Response · Authors · 2025-11-25
>
> 7. **Comment:** "What is the computational cost or time complexity of the proposed training procedure? Since each layer requires retraining previous ones, it appears computationally heavy."
>
> **Response:** Thank you for raising this point. In response to your suggestion, we have conducted an efficiency analysis. The theoretical per-batch computational complexity is $(1+L)L/2$. In practice, for 24-layer ViTs, the additional layer-wise learning introduces an overhead per batch $2.5\times\sim 10\times$. However, this higher per-batch cost does not necessarily make LPA and Adaptive LPA more expensive overall than simultaneous training (Sim), as both methods converge substantially faster and require fewer epochs.
>
> Please refer to Figure 8 in Appendix H (currently included to aid the rebuttal process), where we compare test accuracy versus training epochs for Sim, LPA, and Adaptive LPA on CIFAR-10, CIFAR-100, and Fashion-MNIST. The results show that both LPA and Adaptive LPA reach convergence in fewer epochs than Sim. Specifically, LPA and Adaptive LPA outperform Sim at epochs 70 (99), 44 (50), and 125 (137), achieving test accuracies of 87.32% (87.35%), 61.21% (61.66%), and 93.91% (93.78%), respectively (with values in parentheses corresponding to Adaptive LPA). This demonstrates that training can be stopped much earlier for LPA and Adaptive LPA, effectively compensating for the increased computational overhead.
>
> Furthermore, it can be readily shown that the Layer-wise UAT is defined recursively and continues to hold when multiple layers are treated as a single unit or block. To validate this, we performed additional experiments where LPA was applied to groups of 2, 3, 4, and 6 layers at a time. The outcomes indicate that both the performance and conclusions remain unchanged, and the training time is similar to that of the simultaneous training approach.
>
> From a deployment standpoint, it is important to highlight that LPA and Adaptive LPA enable the creation of multiple usable models from a single training run by allowing users to adjust the model size through pruning later layers. These smaller, tailored models offer much faster inference times compared to the original, full-sized model. Additionally, to the best of our knowledge, we are the first to propose this "one training for N models" approach for residual networks. We believe this added flexibility further compensates for any computational overhead.
>
> Thank you again for your valuable feedback. We will update the manuscript to reflect this analysis.
>
>
> 8. **Comment:** "Prior work (e.g., Greedy Layerwise Learning Can Scale to ImageNet, ICML 2019) already demonstrated over 90% accuracy on CIFAR-10 with layer-wise training, which I believe is the sequential training in this paper. Why does your sequential optimization yield significantly lower accuracy?"
>
> **Response:** We thank you for pointing out the high performance of prior methods. However, there are key differences:
>
> - Greedy Layerwise Learning (ICML 2019) trains one layer at a time, but does not freeze parameters after training, meaning all layers are jointly optimized in later stages.
>
> - In contrast, the Sequential training (Section 4) freezes previously trained layers, ensuring that earlier representations remain stable.
>
> Additionally, our experiments show 94.46% accuracy on CIFAR-10, significantly higher than the 88.3% reported in the paper.
> We also note that the goal of our work is not merely to match existing performance, but to provide a theoretical foundation for understanding and designing efficient training algorithms. The sequential optimization is thus a tool to validate our theory, not the primary objective.

---

> ### Comment · Reviewer_yRZ7 · 2025-11-27
>
> Thank you for the response.
>
> **Further comments**:
>
> The figure 8 in the appendix that shows LPA can achieve similar performance with fewer epochs is interesting. It should be in the main body, along with a discussion on the computational cost, to show more details.
>
> **Remaining concerns**:
> 1. I’m still not convinced why the analysis of UAT in section 3 can inform a new layer-wise training strategy. They may look similar in mathematical form but have fundamentally different objectives, and the layer-wise training strategies have not been demonstrated to implement the idea in section 3, which has nothing to do with loss function at all, as I mentioned earlier.
>
> 2. The authors said prior work (Greedy Layerwise Learning Can Scale to ImageNet, ICML 2019) “...does not freeze parameters after training, meaning all layers are jointly optimized in later stages.” Isn’t this what your method LPA does, training all previous layers (figure 1 (c))?

---

> ### Author Response · Authors · 2025-11-27
>
> We sincerely thank you for your thoughtful feedback.
>
> Yes, we have integrated Fig. 8 to the main body in this revision. In addition, we have updated the experiment on ImageNet to the main body (Line 466).
>
> We have also addressed training, inference, and deployment efficiency in a newly added Section 5.4. In this section, we demonstrate that LPA can not only improve training efficiency but also achieve reductions in both model size and inference time to fit the deployment on low-resource devices (e.g., edge devices).
>
> Furthermore, to investigate the broader applicability of LPA beyond vision models, we conducted a proof-of-concept experiment on a large language model (LLM). This work is presented in a new section in Appendix I, where we demonstrate that the progressive properties of LPA are also evident in large language models.
>
> For your remaining concerns:
>
> 1. **Connection between Section 3 and 4**
>
> The main link between Section 3 and Section 4 is provided by Eq. (13), which emerges as a “by-product” from our investigation of the layer-wise UAT. As a result, Eq. (13) is not directly tied to the UAT itself, which may explain any perceived discrepancy. Nonetheless, Eq. (13) plays a crucial role in Section 4, as it allows us to separately identify and express $f(x_0) - x_0$ and its relationship to the outputs of intermediate layers within the loss function, rather than simply comparing the final output to the ground truth, as is commonly done.
>
> This formulation means that loss reduction can be achieved either through simultaneous learning of all layer parameters or via a multi-step, sequential process (e.g., first reducing
> $f(x_0) - x_0 - G_1(x_0)$, then $f(x_0) - x_0 - G_1(x_0) - G_2(x_1)$, and so on). However, as noted in the main text, sequential learning is not practical. For this reason, we introduce a practical implementation of LPA in Section 4. We hope this explanation clarifies the connection between the two sections.
>
>
> 2. **The authors said prior work (Greedy Layerwise Learning Can Scale to ImageNet, ICML 2019) “...does not freeze parameters after training, meaning all layers are jointly optimized in later stages.” Isn’t this what your method LPA does, training all previous layers (figure 1 (c))?**
>
> No. This is not what the LPA does, but rather the sequential training does. However, LPA is a practical version of sequential training. The differences of LPA from the ICML 2019 papers are: (1) ICML 2019 is specifically tailored for CNNs and does not seek to generalize to the broader family of residual networks; (2) ICML 2019 draws its conclusions from classification tasks where each layer is equipped with an auxiliary classifier, whereas LPA imposes no such requirement and, as demonstrated, can be applied to complex function learning (as shown on synthetic datasets); (3) theoretical analysis of ICML 2019 is primarily concerned with the empirical effectiveness of progressive training in reducing errors, while LPA focuses on establishing the feasibility of progressive training for a wider range of residual network architectures.

---

### Official Review · Reviewer_haSc · 2025-11-01

**Soundness:** 3
**Presentation:** 2
**Contribution:** 3
**Rating:** 4
**Confidence:** 3

**Summary:**

The paper extends the Universal Approximation Theorem (UAT) to modern residual architectures such as ResNets and Transformers. It formulates a layer-wise UAT showing that each residual block satisfies UAT conditions on compact domains, and introduces Layer-wise Progressive Approximation (LPA), a hybrid between sequential and end-to-end training that enforces progressive layer convergence. Theoretical insights are supported by experiments on standard benchmarks (CIFAR-10/100, Fashion-MNIST), where LPA improves accuracy and stability, and an adaptive version enables large compression without accuracy loss.

**Strengths:**

1. The proposed Layer-wise Progressive Approximation (LPA) is intuitively appealing. It aligns with layer-wise pretraining ideas.

2. The paper offers an elegant reformulation of residual layers in a mathematically consistent way, connecting residual mappings to FNN form. This provides a novel, unifying theoretical link between classical approximation theory and modern architectures.

3. The adaptive layer-stopping rule offers a neat way to discover effective depth, leading to large parameter reductions without losing accuracy.

**Weaknesses:**

1. As described in Algorithm 1, a single training epoch appears to involve two full passes over the dataset. The first pass (lines 3-10) is itself computationally intensive, performing L nested backward passes for an L-layer network, resulting in roughly $O(L^2)$ computational complexity per batch. The paper provides no analysis of this overhead (e.g. training time vs. baseline). This makes it impossible to judge the practical efficiency of LPA.

2. LPA is not compared with methods such as Greedy Layer-wise Training [1], Deeply-Supervised Nets [2], and Layer-wise Adaptive Rate Scaling [3]. Without this, it’s unclear if LPA’s gains stem from its theoretical design or from known stabilization effects of progressive optimization.

3. While the results on CIFAR and Fashion-MNIST are strong, these are relatively small-scale datasets. The paper's claims about improved stability and convergence would be far more convincing if demonstrated on a large-scale benchmark such as ImageNet.

4. The writing can be improved. It is verbose and contains several typographical and grammatical errors (e.g., “Universial” instead of Universal, “Archtitectures” instead of Architectures, “Simutaneous” instead of Simultaneous, “patten” instead of pattern).

References

[1] Bengio, Yoshua, et al. "Greedy layer-wise training of deep networks." Advances in neural information processing systems 19 (2006).

[2] Lee, Chen-Yu, et al. "Deeply-supervised nets." Artificial intelligence and statistics. Pmlr, 2015.

[3] You, Yang, Igor Gitman, and Boris Ginsburg. "Large batch training of convolutional networks." arXiv preprint arXiv:1708.03888 (2017).

**Questions:**

Please refer to weaknesses.

---

> ### Author Response · Authors · 2025-11-25
> **Reviewer haSc**
>
> 1. **Comment:** "As described in Algorithm 1, a single training epoch appears to involve two full passes over the dataset. The first pass (lines 3-10) is itself computationally intensive, performing L nested backward passes for an L-layer network, resulting in roughly $O(L^2)$ computational complexity per batch. The paper provides no analysis of this overhead (e.g. training time vs. baseline). This makes it impossible to judge the practical efficiency of LPA."
>
> **Response:** Thank you for bringing this to our attention. Based on your suggestion, we have conducted an efficiency analysis. The theoretical computational complexity per batch is $(1+L)L/2$. In practice, for 24-layer ViTs, the additional layer-wise learning results in a per-batch computational overhead of $2.5\times\sim 10\times$. However, this increased cost per batch does not necessarily make LPA and Adaptive LPA more expensive overall than simultaneous training (Sim), as both methods achieve convergence much earlier and require fewer training epochs than Sim.
>
> Please see Figure 8 in Appendix H (a temporary section for the visual aid of the rebuttal process), where we plot test accuracy against the number of training epochs for Sim, LPA, and Adaptive LPA on CIFAR-10, CIFAR-100, and Fashion-MNIST. The results demonstrate that both LPA and Adaptive LPA converge in fewer epochs than Sim. For instance, LPA and Adaptive LPA outperform the Simultaneous method at epochs 70 (99), 44 (50), and 125 (137), achieving test accuracies of 87.32% (87.35%), 61.21% (61.66%), and 93.91% (93.78%), respectively (with values in parentheses indicating Adaptive LPA). This shows that training can be stopped much earlier for LPA and Adaptive LPA, offsetting the additional computational overhead.
>
> Additionally, it is straightforward to demonstrate that the Layer-wise UAT is recursive and remains valid when multiple layers are grouped together as a single unit (block or layer). To further investigate this, we conducted experiments applying LPA to groups of 2, 3, 4, and 6 layers, respectively. The results show that both performance and conclusions are consistent with previous findings, and the training time is comparable to that of the Sim method.
>
> From a deployment perspective, it is worth emphasizing that LPA and Adaptive LPA make it possible to obtain multiple usable models from a single training run by allowing users to adjust model size through pruning of later layers. These smaller, customized models achieve significantly faster inference times than the original, full-sized model. Furthermore, to our knowledge, we are the first to introduce this "**one training run for N models**" paradigm in the context of residual networks. We believe this flexibility further helps to offset the computational overhead.
>
> Thank you again for your insightful comments. We will update the manuscript to include this analysis.

---

> ### Author Response · Authors · 2025-11-25
>
> 2. **Comment:** “LPA is not compared with methods such as Greedy Layer-wise Training [1], Deeply-Supervised Nets [2], and Layer-wise Adaptive Rate Scaling [3]. Without this, it’s unclear if LPA’s gains stem from its theoretical design or from known stabilization effects of progressive optimization.”
>
> **Response:** Thank you for sharing these valuable references. We were not aware of them during the preparation of our manuscript. We have now compared our method’s performance with these approaches (see results below) and analyzed their similarities and differences, which will be incorporated into the revised version. Additionally, we identified another recent extension of [1], namely [4] "Greedy Layerwise Learning Can Scale to ImageNet" (ICML 2019).
>
> Both Greedy Layer-wise Training [1] and Greedy Layer-wise Learning [4] utilize a layer-wise training approach with a shape similar to LPA. However, the lack of available source code prevented us from investigating further implementation details. Based on the descriptions in the papers, we observe the following differences: (1) Greedy Layer-wise Training [1] is a practical method for representation learning and adopts a strategy akin to the sequential learning paradigm we discuss; (2) Greedy Layer-wise Learning [4] is specifically tailored for CNNs and does not seek to generalize to the broader family of residual networks; (3) [4] draws its conclusions from classification tasks where each layer is equipped with an auxiliary classifier, whereas LPA imposes no such requirement and, as demonstrated, can be applied to complex function learning (see results on synthetic datasets); (4) their theoretical analysis is primarily concerned with the empirical effectiveness of progressive training in reducing errors, while LPA focuses on establishing the feasibility of progressive training for a wider range of residual network architectures.
>
> Deeply-Supervised Nets [2] shares some similarity to LPA in that it introduces intermediate loss signals to facilitate training. However, this work primarily focuses on practical aspects.
>
> With respect to Layer-wise Adaptive Rate Scaling [3], it is mainly an engineering method used for large-batch training, without introducing structural modifications to the network or employing layer-wise supervision. Since it is described in a technical report with limited experimental details, we were unable to perform both quantitative and qualitative comparisons. Nevertheless, we recognize its relevance to optimization stability and will cite it in the related work section of the revised manuscript.
>
> | Method | CIFAR-10 | ImageNet |
> |--------|----------|----------|
> | Greedy Layerwise Learning Can Scale to ImageNet, ICML 2019 | 91.7 | 69.7 |
> | Deeply-Supervised Nets | 91.78 | - |
> | Ours | 93.51 | 78.58 |
>
> Note: We use the ICML 2019 result instead of the original [1] because the latter paper performance on non-standard or outdated benchmarks, making direct comparison difficult. The ICML 2019 work implements a scalable version of the same core idea and is more comparable to our setting.
>
> 3. **Comment:** "While the results on CIFAR and Fashion-MNIST are strong, these are relatively small-scale datasets. The paper's claims about improved stability and convergence would be far more convincing if demonstrated on a large-scale benchmark such as ImageNet."
>
> **Response:**  Thank you for your insightful suggestion. We recognize the importance of scalability and have performed experiments on ImageNet using a 12-layer ViT model. The results, presented below, are consistent with our previous findings on CIFAR-10/100 and Fashion-MNIST: models trained with LPA demonstrate progressive learning, with earlier layers attaining performance comparable to those trained under the simultaneous paradigm. We also intend to extend our evaluation to a 24-layer ViT and will include these results in the revised manuscript.
>
> | Layer | 2     | 4     | 6     | 8     | 10    | 12    |
> |-------|-------|-------|-------|-------|-------|-------|
> | Sim   | 0.152 | 0.336 | 0.964 | 9.558 | 48.428| 77.752|
> | LPA   | 33.196| 56.432| 67.660| 73.636| 77.344| 78.586|
>
> 4. **Comment:** "The writing can be improved. It is verbose and contains several typographical and grammatical errors (e.g., “Universial” instead of Universal, “Archtitectures” instead of Architectures, “Simutaneous” instead of Simultaneous, “patten” instead of pattern)."
>
> **Response:** Thank you for your careful review and for pointing out the typos and errors. We will thoroughly proofread the paper and make improvements to the writing.

---

> ### Comment · Reviewer_haSc · 2025-11-27
>
> Thank you for the detailed response, particularly the efficiency analysis and the preliminary ImageNet results. While I appreciate these additions, the confirmed $O(L^2)$ computational complexity and significant per-batch overhead ($2.5\times \sim 10\times$) remain critical barriers to practical scalability that are not fully mitigated by the reduced training epochs. I will therefore maintain my current rating.

---

> > ### Author Response · Authors · 2025-11-27
> >
> > Thank you for your prompt response and for clarifying your main concern regarding training efficiency. However, we believe it is more appropriate to evaluate training, inference, and deployment efficiency as well as accuracy together, rather than focusing solely on training time. We would like to clarify this by highlighting several common practical scenarios in which organizations balance these factors:
> >
> > **Case 1: Prioritizing Accuracy and Inference Efficiency**
> >
> > Many organizations are primarily concerned with model accuracy and inference speed, and are willing to accept additional training overhead. Our results show that LPA not only improves final layer performance but also often enhances early layer accuracy, while maintaining equal or reduced inference time.
> >
> > **Case 2: Prioritizing Inference Time and Deployment Constraints**
> >
> > Some organizations need to deploy models on resource-constrained devices (such as edge devices) where inference time and model size are critical. LPA supports this need by enabling substantial model compression. For example, our experiments demonstrate that a model compressed to one-fourth the size of the original 24-layer ViT can achieve a 4x reduction in inference time, making deployment on low-resource devices possible. In addition, we have done experiment to show that a 6-layer model reduced by LPA can be deployed on the Raspberry Pi Zero, something unattainable with the full-sized model.
> >
> > **Case 3: Prioritizing Training Time**
> >
> > In situations where training time is the main concern (as you mentioned), LPA allows for flexibility by adjusting the number of training epochs or the number of layers grouped per LPA unit. Our experiments show that reducing these parameters can significantly decrease training time while still achieving competitive accuracy, even outperforming the baseline in some cases. For example, with a 70 training epochs and the number of layers in a LPA unit at 12 on Cifar-10 it shows a 30% of training time reduction compared to a 24-ViT of 300 epochs to reach it performance convergence. Furthermore, the accuracy has been improved to 88% (compared to 86% of the 24-ViT).
> >
> > In summary, different deployment scenarios require balancing these factors, and we believe LPA offers valuable flexibility to meet a variety of practical needs. We hope these advantages are appreciated.

---

> ### Author Response · Authors · 2025-11-28
>
> Thank you very much for your valuable feedback.
>
> We would like to update that we have updated the PDF, correcting all identified typos and errors. In addition, we have added two new sections (Section 5.4 and Appendix I), with all revisions clearly highlighted for your convenience. The updated version is now available in the submission.
>
> Specifically, in response to the important question of LPA’s scalability on large-scale, real-world datasets, we have included new experiments on ImageNet (see Line 466). These results demonstrate that the progressive approximation property of LPA continues to hold, even in this more demanding context, thereby underscoring the robustness and practical value of our framework.
>
> Section 5.4 now discusses training, inference, and deployment efficiency. In this section, we show that LPA not only improves training efficiency, but also reduces model size and inference time, which facilitates deployment on low-resource devices such as edge hardware.
>
> Additionally, to further explore the generalizability of LPA beyond vision models, we conducted a proof-of-concept experiment on a large language model (LLM). The findings are presented in the new Appendix I, where we show that the progressive nature of LPA is also present in large language models.

---

### Official Review · Reviewer_QEy9 · 2025-11-01

**Soundness:** 2
**Presentation:** 2
**Contribution:** 3
**Rating:** 4
**Confidence:** 4

**Summary:**

The paper extends the universal approximation theorem (UAT) to residual networks and thus other architectures such as convolutional networks Transformer architectures. The extension is done by recognizing that each residual block itself is a one-hidden-layer network and thus we can apply UAT in a suitable fashion to show the universal approximation. Using this concept, a layer-wise progressive approximation (LPA) is proposed to train a residual network layer by layer. Two groups of experiments are conducted. One for verifying the approximation ability on synthetic datasets and the other one for real image recognition datasets. On the approximation side, LPA shows order of magnitude improvements over end-to-end training and sequential optimization. On the image recognition side, LPA shows better performance than the other two baselines on CIFAR-10, CIFAR-100, and Fashion-MNIST.

**Strengths:**

The paper is well-written with good clarity. Exploring different options for training is a very important research avenue and this paper compares three different training techniques on both approximation and generalization tasks, which is interesting and insightful.

**Weaknesses:**

It is trivial that a residual network can satisfy universal approximation and thus for other works due to the use of MLP or an equivalent form. I believe LPA has good approximation. However, I am not convinced that LPA has better generalization than end-to-end training due to the limited experiments. The CIFAR-10/100 and Fashion-MNIST are very small datasets. On the other hand, like proving UAT, it is also important to prove some generalization bounds for LPA if possible. If not, what is the main difficulty? From an optimization viewpoint, a reader would expect some guarantees or results on the quality of the solution or the optimization landscape. For example, can we show that by LPA, the network always outperforms certain strong predictors? Some related works are also missing in this aspect. There are some nice properties for the optimization landscape of a residual network. See the following references for instance.

[1] Chen, Kuan-Lin, Ching-Hua Lee, Harinath Garudadri, and Bhaskar D. Rao. "ResNEsts and DenseNEsts: Block-based DNN models with improved representation guarantees." Advances in neural information processing systems 34 (2021).

[2] Yun, Chulhee, Suvrit Sra, and Ali Jadbabaie. "Are deep ResNets provably better than linear predictors?." Advances in Neural Information Processing Systems 32 (2019).

**Questions:**

1.	In all the experiments, did we apply any learning rate scheduling to make sure different training strategies get their best learning rate schedule?
2.	The datasets used are very small. Have you tried ImageNet or some real datasets other than image recognition?
3.	In the ResNet architecture, there is a nonlinearity before the last linear layer. Eq. (10) does not include that. Is this a simplified ResNet?
4.	Do we use batch normalization in the experiments?
5.	How to theoretically study the generalization ability of LPA? Is it possible to derive some generalization bounds for it?

---

> ### Author Response · Authors · 2025-11-25
> **Reviewer QEy9**
>
> Thank you for your thoughtful and insightful comments. We appreciate your constructive feedback, which has helped us clarify the scope and strengths of our work. Below are our point-by-point responses:
> 1. **Comment:** "In all the experiments, did we apply any learning rate scheduling to make sure different training strategies get their best learning rate schedule? "
>
> **Response:** Thank you for your suggestion. In our initial study, we used a fixed empirical learning rate of 0.0001 and did not examine its sensitivity. Following your advice, we have now evaluated the learning rate on CIFAR-10 by varying it within the range [0.00005, 0.0005]. The results are presented below and Figure 9 in the Appendix H (a temporary section for the visual aid of the rebuttal process).
>
> The results suggest that performance is generally insensitive to the choice of learning rate, although the fixed rate used in our experiments was not optimal. Thank you again for this valuable suggestion. We will include this update in the revised manuscript.
>
>
> | Learning Rate | 0.00005 | 0.00008 | 0.0001 | 0.0002 | 0.0003 | 0.0004 | 0.0005 |
> |---------------|--------|--------|--------|--------|--------|--------|--------|
> | Acc           | 88.32  | 89.25  | 89.37  | 90.13  | 90.92  | 90.14  | 89.87  |
>
>
> 2. **Comment:** "The datasets used are very small. Have you tried ImageNet or some real datasets other than image recognition?"
>
> **Response:** Thank you for your valuable suggestion. We recognize the importance of scalability and have conducted experiments on ImageNet using a 12-layer ViT model. The results, presented below, are consistent with our findings on CIFAR-10/100 and Fashion-MNIST: models trained with LPA exhibit a progressive property, with earlier layers achieving performance comparable to those trained using the simultaneous paradigm. We plan to further evaluate the approach using a 24-layer ViT and will update the results in the revised manuscript.
>
> | Layer | 2     | 4     | 6     | 8     | 10    | 12    |
> |-------|-------|-------|-------|-------|-------|-------|
> | Sim   | 0.152 | 0.336 | 0.964 | 9.558 | 48.428| 77.752|
> | LPA   | 33.196| 56.432| 67.660| 73.636| 77.344| 78.586|
>
> 3. **Comment:** "In the ResNet architecture, there is a nonlinearity before the last linear layer. Eq. (10) does not include that. Is this a simplified ResNet?"
>
> **Response:** We utilized the standard ResNet implementation provided in the official repositories, without any simplifications. All experimental setups faithfully retain the original ResNet architecture, including batch normalization and activation functions.
> Equation (10) in our paper serves as an abstract representation of the network output, rather than a precise architectural blueprint. This level of abstraction is intended to capture the general structure shared by residual networks, such as ResNets and Transformers, thereby making LPA broadly applicable to these architectures. Thank you for highlighting this point. We will provide further clarification in the revised manuscript.
>
> 4. **Comment:** "Do we use batch normalization in the experiments?"
>
> **Response:** Yes, batch normalization was used in all experiments.
>
> 5. **Comment:** "How to theoretically study the generalization ability of LPA? Is it possible to derive some generalization bounds for it?"
>
> **Response:** Thank you for bringing this to our attention. In Section 3.2, we show that LPA can be generalized to all residual networks whose layers can be formulated as in Eq. (3), and we further clarify the input-output mapping conditions in Appendix G (as noted by Reviewer QEy9). However, we are cautious not to claim universal generalizability for all residual networks, as this would be overreaching. For each specific residual network model, the applicability of the layer representation and the input-output conditions must be assessed individually. This point is demonstrated in the Appendix, where extra steps are necessary for models like Transformers. As a result, we have focused on establishing generalization conditions rather than strict bounds. We appreciate you highlighting this important issue, and we plan to investigate it further in future work.
>
> 6. **Comment:** "See the following references for instance."
>
> [1] Chen, Kuan-Lin, Ching-Hua Lee, Harinath Garudadri, and Bhaskar D. Rao. "ResNEsts and DenseNEsts: Block-based DNN models with improved representation guarantees." Advances in neural information processing systems 34 (2021).
>
> [2] Yun, Chulhee, Suvrit Sra, and Ali Jadbabaie. "Are deep ResNets provably better than linear predictors?." Advances in Neural Information Processing Systems 32 (2019).”
>
> **Response:** Thank you for sharing these two valuable references. We will incorporate them into the related work section and discuss how they relate to our study.

---

> ### Author Response · Authors · 2025-11-28
>
> We sincerely appreciate your thoughtful feedback.
>
> We are writing to update that we have revised the PDF to correct typos and errors. Additionally, we have included two new sections (Section 5.4 and Appendix I), with all changes clearly highlighted. The updated version is available in the revisions.
>
> In particular, acknowledging the importance of LPA’s scalability to large-scale, real-world datasets, we have added an ImageNet experiment (see Line 466). The results confirm that LPA’s progressive approximation property remains valid in this more challenging setting, further demonstrating the robustness and practical relevance of our approach.
>
> We have also addressed training, inference, and deployment efficiency in the newly added Section 5.4. Here, we show that LPA can enhance training efficiency and reduce both model size and inference time, making it suitable for deployment on resource-constrained devices such as edge devices.
>
> Finally, to explore LPA’s applicability beyond vision models, we conducted a proof-of-concept experiment on a large language model (LLM). This work is detailed in the new Appendix I, where we demonstrate that LPA’s progressive properties are also observed in large language models.

---

### Official Review · Reviewer_dNkd · 2025-11-01

**Soundness:** 3
**Presentation:** 3
**Contribution:** 2
**Rating:** 6
**Confidence:** 3

**Summary:**

The paper revisits the universal approximation theorem (UAT) from a layer-wise perspective, showing that residual and Transformer architectures can, in principle, achieve universal approximation through a recursive composition of continuous mappings. It further proposes a practical Layer-wise Progressive Approximation (LPA) algorithm inspired by this theoretical view. Overall, the theoretical framing is interesting and the exposition is clear.

**Strengths:**

1. The paper is readable, with equations and intuition laid out clearly.
2. The idea of connecting layer-wise residual updates with UAT is elegant and pedagogically valuable.
3. The paper provides a new viewpoint for understanding how deep residual structures build representations progressively.
4. Experiments, though small-scale, are well-organized and confirm the feasibility of the proposed algorithm.

**Weaknesses:**

1. The paper needs to be polished. Several typos exist, and there are notations that could be made clearer.
2. I am not sure how strong the claim that 'prior work struggled to obtain generalizable conclusions because it attempted to model entire multi-layer architectures at once' is, because at least for ResNet, the UAT conclusion from a typical FNN could be easily adapted with minimal additional efforts.
3. Line 155, in the derivation from single-layer RN to two-layer RN, the argument implicitly replaces a target $f_{2}(x_0)$ (a function of $x_0$) to $G_2(x_1)$ (a function of $x_1$.) Even though the authors state that $x_1$ is fixed from $G_1^*$, it holds only when there's an explicit mapping or invertibility assumption between $x_0$ and $x_1$ so that the target function can be reparameterized in the $x_1$-domain. Interestingly, though it has been proved in Appendix G (Factorization Continuity Theorem), the authors do not mention it in the main text. I wonder why?

Just a minor suggestion: maybe the authors can consider compressing the image size in the paper. The current PDF size is 30MB, which is a bit large.
Several typos in the paper:
1. Line 80-90, should be 'various' instead of 'verious';
2. Line 134-135, 'continuous' instead of 'contineous';
3. A FNN/FFN -> An FNN/FFN;

**Questions:**

Please refer to the weakness.

---

> ### Author Response · Authors · 2025-11-25
> **Reviewer dNkd**
>
> We sincerely appreciate your thorough review of our manuscript and the insightful feedback you have provided. Your suggestions have been extremely helpful in enhancing the quality of our work. Please find our detailed, point-by-point responses to your comments below:
>
> 1. **Comment:** “The paper needs to be polished. Several typos exist, and there are notations that could be made clearer.”
>
> **Response:** Thank you for the suggestion. We will improve in the revision.
>
>
> 2. **Comment:** "I am not sure how strong the claim that 'prior work struggled to obtain generalizable conclusions because it attempted to model entire multi-layer architectures at once' is, because at least for ResNet, the UAT conclusion from a typical FNN could be easily adapted with minimal additional efforts."
>
> **Response:** Thank you for pointing out this inaccurate wording, particularly the use of “struggled.” A more accurate statement is that previous work has primarily focused on architecture-specific proofs, and there has not yet been an attempt to draw conclusions that are generalizable to the broader family of residual networks. We will rephrase in the revision.
>
>
> 3. **Comment:** "Line 155, in the derivation from single-layer RN to two-layer RN, … the authors state that $x_1$ is fixed from $G_1^*$, it holds only when there's an explicit mapping or invertibility assumption between $x_0$ and $x_1$ so that the target function can be reparameterized in the $x_1$-domain. Interestingly, though it has been proved in Appendix G (Factorization Continuity Theorem), the authors do not mention it in the main text. I wonder why?"
>
> **Response:**  Thank you for pointing out the missing reference. The citation to Appendix G was inadvertently removed while we were condensing the manuscript to meet the page limit. We will restore this reference in the revised version.
>
>
> 4. **Comment:** "Just a minor suggestion: maybe the authors can consider compressing the image size in the paper. The current PDF size is 30MB, which is a bit large. Several typos in the paper…"
>
> **Response:** Thank you very much for the suggestion. We will compress the images and correct the typos in the revision.

---

> > ### Comment · Reviewer_dNkd · 2025-11-27
> >
> > Thank you for your response and clarifications. I appreciate that the authors addressed each point and plan to fix the wording, restore the missing theoretical proof reference to Appendix G, and polish typos and formatting issues in the revision. These changes will certainly improve the readability and precision of the paper. I look forward to seeing a more polished version in future iterations! I will keep my original score of 6.

---

> ### Author Response · Authors · 2025-11-27
>
> We sincerely thank you for your thoughtful feedback. In addition to carefully correcting all typos as requested (highlighted in yellow for ease of review) and compressing the PDF to approximately 17 MB, we have taken this opportunity to further strengthen the empirical foundation of our work.
>
> Specifically, recognizing that the scalability of LPA to large-scale real-world datasets is a natural and important question, we have added an experiment on ImageNet (Line 466). The results confirm that the progressive approximation property of LPA continues to hold even in this more challenging setting, thereby reinforcing the robustness and practical relevance of our framework.
>
> We have also addressed training, inference, and deployment efficiency in a newly added Section 5.4. In this section, we demonstrate that LPA can not only improve training efficiency but also achieve reductions in both model size and inference time to fit the deployment on low-resource devices (e.g., edge devices).
>
> Furthermore, to investigate the broader applicability of LPA beyond vision models, we conducted a proof-of-concept experiment on a large language model (LLM). This work is presented in a new section in Appendix I, where we demonstrate that the progressive properties of LPA are also evident in large language models.

---

### Meta-Review · Area_Chair_nznt · 2025-12-08

**Summary:**

The paper proposes a layer-wise universal approximation view for residual networks and a progressive optimization scheme (LPA), and shows some empirical gains over standard end-to-end training. However, as several reviewers pointed out, the theoretical contribution appears incremental over standard UAT for FNNs, the connection between the theory and the proposed training algorithm is loose, and the training cost is significantly higher. Given the almost non-existent rebuttal, I agree with the reviewers that the work is not yet ready for main conference acceptance and recommend rejection.

**Reviewer Concerns:**

I find the reviewers’ comments on limited theoretical novelty, the weak formal link between the approximation results and the LPA optimization algorithm, and the O(L²)-like training cost/scalability issues to be well-founded and largely unaddressed by the rebuttal. Writing/typo issues and missing baselines/large-scale experiments could in principle be fixed in a revision, but even if these were improved, the core concerns about novelty and practicality would remain. Therefore, I do not believe any major reviewer concern has been convincingly resolved.

**Reviewer Scores:**

For the more positive reviewer, a future revision that improves clarity and adds stronger experiments might slightly raise their score, but under the current version and rebuttal I expect them to stay at most borderline reject. For the more critical reviewers who focused on theoretical originality and scalability, I see no reason their scores would improve, as their main concerns were not meaningfully addressed. Overall, I expect reviewers’ scores to remain essentially unchanged, leading to a consensus toward rejection.

---

### Decision · Program_Chairs · 2026-01-26

Reject